# Trichuris muris infection drives cell-intrinsic IL4R alpha independent colonic RELMα+ macrophages

**Ruth Forman**[1]*, **Larisa Logunova**[1], **Hannah Smith**[1], **Kelly Wemyss**[1], **Iris Mair**[1], **Louis Boon**[2], **Judith E. Allen**[1], **Werner Muller**[1], **Joanne L. Pennock**[1], **Kathryn J. Else**[1]*

**1** Lydia Becker Institute of Immunology and Inflammation, Faculty of Biology, Medicine and Health, The University of Manchester, Manchester Academic Health Science Centre, Manchester, United Kingdom, **2** Polpharma Biologics, Utrecht, The Netherlands

* ruth.forman@manchester.ac.uk (RF); Kathryn.else@manchester.ac.uk (KJE)

**Data Availability Statement:** All relevant data are within the manuscript and its Supporting Information files.

## Abstract

The intestinal nematode parasite *Trichuris muris* dwells in the caecum and proximal colon driving an acute resolving intestinal inflammation dominated by the presence of macrophages. Notably, these macrophages are characterised by their expression of RELMα during the resolution phase of the infection. The RELMα+ macrophage phenotype associates with the presence of alternatively activated macrophages and work in other model systems has demonstrated that the balance of classically and alternatively activated macrophages is critically important in enabling the resolution of inflammation. Moreover, in the context of type 2 immunity, RELMα+ alternatively activated macrophages are associated with the activation of macrophages via the IL4Rα. Despite a breadth of inflammatory pathologies associated with the large intestine, including those that accompany parasitic infection, it is not known how colonic macrophages are activated towards an alternatively activated phenotype. Here, we address this important knowledge gap by using *Trichuris muris* infection, in combination with transgenic mice (IL4Rαfl/fl.CX3CR1Cre) and IL4Rα-deficient/wild-type mixed bone marrow chimaeras. We make the unexpected finding that education of colonic macrophages towards a RELMα+, alternatively activated macrophage phenotype during *T. muris* infection does not require IL4Rα expression on macrophages. Further, this independence is maintained even when the mice are treated with an anti-IFNγ antibody during infection to create a strongly polarised Th2 environment. In contrast to RELMα, PD-L2 expression on macrophages post infection was dependent on IL4Rα signalling in the macrophages. These novel data sets are important, revealing a surprising cell-intrinsic IL4R alpha independence of the colonic RELMα+ alternatively activated macrophage during *Trichuris muris* infection.

## Author summary

Infection of mice with *Trichuris muris*, a whipworm parasite results in inflammation of the large intestine. Inflammation is temporary; once the parasite has been cleared, damage

**Funding:** This work was supported by a Medical Research Council (https://mrc.ukri.org/funding/) grant (MR/N022661/1) awarded to KJE with additional support from a Wellcome Trust (106898/A/15/Z) grant to JEA. The funders had no role in study design, data collection and analysis, decision to publish, or preparation of the manuscript.

**Competing interests:** The authors have declared that no competing interests exist.

to the intestinal tissue heals. During inflammation white blood cells move in to the gut tissue. These cells are dominated by a cell type called the macrophage. Macrophages which accumulate in the intestine post-infection express a protein called RELMα. These RELMα-expressing macrophages are thought to help resolve inflammation and have traditionally been associated with IL-4 and IL-13-driven activation. We set out to determine whether the macrophages which emerge during *T. muris* infection need to respond to IL-4 and/or IL-13 in order to express RELMα. We did this by creating a transgenic mouse where the common IL4Rα chain of the IL-4 and IL-13 receptor was absent from macrophages. To our surprise, macrophages were able to express RELMα regardless of whether the macrophage could or could not respond to IL-14/IL-13. This new knowledge is important as in some inflammatory conditions, treatments seeking to encourage alternatively activated macrophages have been proposed. Such treatments require an understanding of both the important and the redundant signals as well as recognition that activating signals may be disparate in different tissue environments.

## Introduction

*Trichuris muris (T. muris)* is a natural parasite of mice that resides in the cecum and proximal colon. Immunity to *T. muris* infection is driven by a type two immune response, characterised by the production of interleukin (IL)4, IL5, IL9 and IL13 and accompanied by the generation of a parasite specific IgG1 response. The exact mechanisms these type 2 effector responses use to drive the expulsion of the parasite is only partially known but goblet cells, mucin production [1,2], increased intestinal muscle contractility driven by IL9 [3] and increased epithelial cell turnover driven by IL13 [4] have all been shown to aid expulsion.

As *T. muris* burrows through the epithelial cells within the large intestine, maintenance of the barrier function and control of inflammation of the large intestine is imperative to protect the host from sepsis. Whilst infection drives recruitment in to the gut of variety of leukocytes, macrophages are the dominant cell type [5,6]. Macrophages are the most common mononuclear phagocyte in the healthy intestinal lamina propria and act as vital immune sentinels, able to sense and respond to changes within the gut. Indeed, macrophage dysfunction results in an excessive inflammatory response and a failure to resolve inflammation [7]. In contrast to, for example, Inflammatory Bowel Disease (IBD), where classically activated macrophages predominate, alternatively activated macrophages are a feature of many helminth infections [8]. Consistent with this, the macrophages recruited during *T. muris* infection adopt an alternatively activated phenotype and are marked by the expression of RELMα. These RELMα[+] macrophages persist post-expulsion, as intestinal inflammation resolves and the gut architecture is restored [6]. The abundance, phenotype and kinetics of emergence of the RELMα[+] macrophage during whipworm has lead to their being proposed to be important in restraining potentially damaging immunopathology as well as a playing a role in remodelling and repair [6]. However, despite the abundance of these cells within the intestine post-infection a full understanding of the *in vivo* activating signals, which enable a balance of macrophage subpopulations to emerge and regulate inflammatory processes, is lacking. This is essential if we are to understand the mechanisms of chronic inflammation and develop targeted therapies to improve disease outcome [9]. Given that a Type 2 immune response and the Type 2 cytokines IL4 and IL13 are vital for the expulsion of *T. muris*, we aimed to identify the role these cytokines play in shaping the intestinal macrophage response seen post infection.

Distinct macrophage polarization was first described by Gordon and colleagues in the context of classical activation, driven by IFNγ and LPS; and alternative activation, driven by IL4

and IL13 [10,11]. It later became clear that many different stimuli could lead to an alternatively activated macrophage phenotype, collectively referred to as M2. The term M2 thus embraces a wide spectrum of alternatively activated macrophage states and consequently has lost some utility. Murray and colleagues [12] attempted to bring clarity to M2 nomenclature describing a set of markers associated with alternatively activated macrophages activated *in vitro* by IL-4. These macrophages, coined M(IL4), are identified as RELMα$^+$ Ym1$^+$ CD206$^+$ and PD-L2$^+$ macrophages. Many laboratories have demonstrated that these markers are also IL4Rα dependent *in vivo* and these have become reliable indicators of M(IL4), especially in the context of type 2 inflammation and helminth infections. In particular, studies utilising the small intestinal nematodes *Heligmosomoides polygyrus* and *Nippostrongylus brasiliensis* have shown that RELMα$^+$ Ym1$^+$ CD206$^+$ macrophages infiltrate the small intestine in an IL4/IL13 dependent manner [13,14]. Similarly to these intestinal nematodes, it has been reported that the filarial nematode *Brugia malayi* drives an IL4Rα dependent RELMα$^+$ Ym1$^+$ Arg-1$^+$ macrophage population within the peritoneal cavity [15]. More recently, in the absence of any infection, IL4Rα -independent pathways of Ym1 and RELMα expression have been described in lung macrophages [16]. Such steady state IL4Rα–independence was subsequently shown not to be limited to the lung, with both IL4Rα and the downstream transcription factor STAT6 dispensable for RELMα expression in steady state peritoneal macrophages [17]. Beyond the steady state, there is also a precedent in the literature for the induction of RELMα on macrophages to be partially, but not entirely, IL4Rα independent in the context of lung and liver inflammation, following *Nippostrongylus brasiliensis* and *Schistosoma mansoni* infection respectively [16,18]. However, it remains unknown whether intestinal macrophages depend on IL4Rα signalling for their education during helminth infection. Attempts to address the importance of the IL4Rα in the context of colonic macrophages have been limited by the Cre driver employed to delete the IL4Rα from macrophages and the markers used to define the M2 phenotype. Thus, there remains no clear exploration of the role of IL4Rα in the education of colonic macrophages during either steady state or in an inflamed gut environment during helminth infection. This represents a surprising gap in our knowledge, given that it is essential information underpinning studies striving to enrich for alternatively activated macrophages in pro-inflammatory settings such as inflammatory bowel disease.

We set out to evaluate the role of IL4Rα in educating macrophages during *T. muris* infection, given the recognised importance of IL4 and IL13 in promoting alternatively activated macrophages in other inflamed tissues [12,16,19–21]. To our surprise, using a combination of *in vivo* approaches, including transgenic mice (IL4Rαfl/fl.CX3CR1Cre) and mixed bone marrow chimaeras, we found that during infection driven inflammation, IL4Rα on colonic macrophages was completely dispensable for the emergence of RELMα$^+$Ym1$^+$ macrophages. This work is an important addition in our understanding of chronic intestinal inflammation and impacts upon strategies to treat these conditions. In particular, these findings highlight that targeting the IL4R to promote alternatively activated macrophages may not be efficacious. Our work also demonstrates the need to understand more about macrophage biology *in vivo* in physiological contexts, and to corroborate *in vitro* observations with *in vivo* evidence investigating macrophages at sites of inflammation.

## Results and discussion

### Global gene expression analyses of the *T. muris* associated intestinal macrophage reveals a broad alternatively activated phenotype

Infection with the gut dwelling nematode parasite *T. muris* is associated with the development of alternatively activated RELMα$^+$ macrophages [6]. To characterise these macrophages we

performed bulk RNA seq on monocytes and macrophages isolated from the caecum and colon of infected C57BL/6 mice. Monocyte and macrophage populations were defined based on the previously described monocyte/macrophage waterfall where Ly6C[hi] blood monocytes enter the intestine (P1), progressively upregulate surface MHC II expression (P2) and down regulate Ly6C expression to become mature gut macrophages (P3) [22–25]. Cells were pooled from two mice, sorted using FACS and isolated into two subsets based on surface expression of the marker CD206, chosen based on good co-expression with intracellular RELMα post *Trichuris* infection (S1A Fig). Counts per gene demonstrated good quality RNA seq reads (S1B Fig) and principal component analysis (PCA) analyses of the two monocyte / macrophage subsets revealed a detectable influence of animal sample on the gene expression profile (22%) but this was significantly weaker than the difference attributed to expression of CD206 (58% of the sample variance; S1C Fig). In addition, *Mrc1*, the gene encoding CD206 and *Retnla* the gene encoding RELMα, were significantly upregulated in the CD206 positive population, validating the sorting strategy (S1D Fig). Importantly, volcano plot analysis of the differential gene expression between the 2 macrophage populations revealed that the CD206[+] macrophage profile associated with genes such as *Retnla*, *Mrc1*, *Ccl8*, *CD163*, *Ccl6*, and *Gas6* which have all been described to be upregulated in alternatively activated macrophages [26] (S1D Fig). In addition, IPA analysis demonstrated an enrichment in our dataset of genes associated with the inflammatory response ($p = 6.69 \times 10^{-20}$), inflammation of the gastrointestinal tract ($p = 6.76 \times 10^{-23}$) and inflammation of the large intestine ($p = 4.54 \times 10^{-18}$). Because the transcriptional landscape fits the canonical alternatively activated macrophage phenotype [19,26], this suggests data gathered utilising *T. muris* driven inflammation is translatable to other intestinal inflammatory conditions. Further, given that recent work has suggested that not all RELMα detected in lung macrophages is actually macrophage derived [17], with a small proportion (10%) of alveolar macrophages able to acquire cytosolic proteins e.g. RELMα from adjacent pneumocytes in the lung alveoli, the RNAseq analysis (S1C Fig) is important in demonstrating that Retnla is highly expressed at the mRNA level in intestinal macrophages post *T. muris* infection.

## IL4Rαfl/fl.CX3CR1Cre+ macrophages are un-responsive to IL4 stimulation

To determine the *in vivo* role of IL4/IL13 in macrophage education we crossed CX3CR1Cre mice [27] with IL4Rαfl/fl mice [28] to generate IL4Rαfl/fl.CX3CR1Cre mice. Macrophages from these IL4Rαfl/fl.CX3CR1Cre+ mice lack the IL4Rα chain, and therefore are unable to respond to IL4 and IL13. In contrast, littermate controls (IL4Rαfl/fl.CX3CR1Cre-) remain IL4/IL13 responsive. The Cre driver was chosen as it is known to efficiently delete floxed alleles in all intestinal macrophages [27]. This is important, as historically, determining the function of genes in macrophages has been confounded by inefficacies in deletion of floxed genes when using the LysM Cre driver [29]. The inefficient deletion of genes by LysM Cre is observed in subpopulations of macrophages which express low levels of Lyz2. Therefore, whilst LysM Cre is able to efficiently delete IL4Rα under steady state conditions, during inflammation a large proportion of macrophages retain IL4Rα expression [29,30]. In order to confirm the efficient deletion of the IL4Rα in CX3CR1Cre[+] colonic macrophages, we injected mice intraperitoneally with a recombinant IL4 complex (IL4c). Under uninflamed steady state conditions, RELMα expression in naïve macrophages from the peritoneal cavity (Fig 1A and 1B), gut (Fig 1C and 1D) and liver (Fig 1E and 1F) all showed no dependency on IL4Rα in keeping with the literature from lung [16] and peritoneal macrophages [17]. Using the IL4Rαfl/fl.CX3CR1Cre mice, and in accordance with data previously reported with global IL4Rα deficient mice [31], IL4c drove an IL4Rα-dependent activation of F4/80[hi] macrophages within the peritoneal

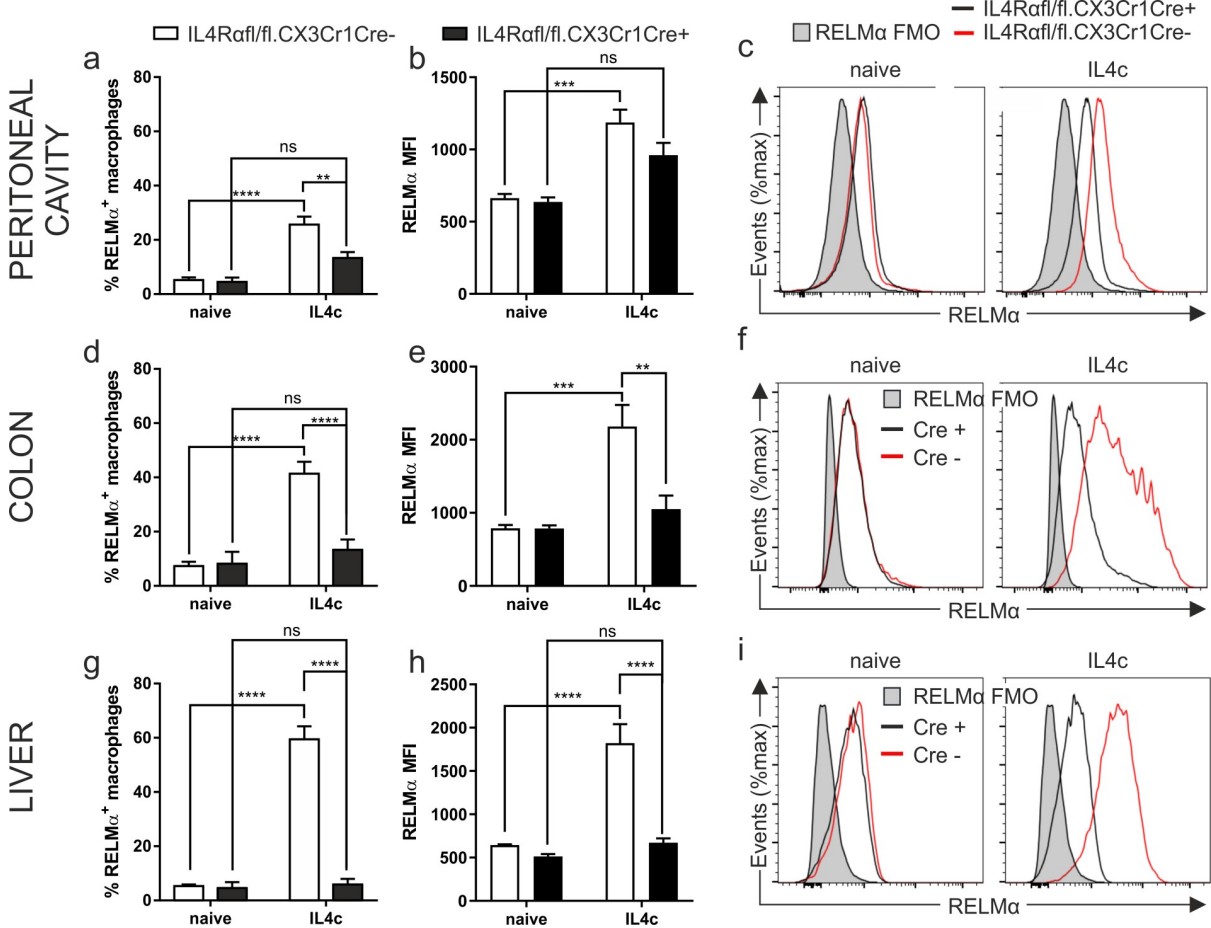

**Fig 1. IL4Rαfl/fl.CX3CR1Cre⁺ macrophages are unresponsive to IL4 signalling.** IL4Rαfl/fl.CX3CR1Cre+ or IL4Rαfl/fl.CX3CR1Cre- mice were injected intraperitoneally with recombinant IL4 complex and expression of RELMα analysed in the (a,b) peritoneal cavity, (d,e) colon and (g) liver 48hours later. Representative histograms show RELMα expression in the IL4Rαfl/fl.CX3CR1Cre- (red) or IL4Rαfl/fl.CX3CR1Cre+ (black) mice in the (c) peritoneal cavity, (f) colon and (i) liver compared to the FMO control. Data are combined from two independent experiments (naïve; n = 3–7, IL4c; n = 6). Error bars show means ± SEM. Statistical comparisons were performed with a two-way ANOVA with post-hoc Bonferroni's multiple comparison test where appropriate: *$p<0.05$, **$p<0.01$, ***$p<0.001$, ****$p< 0.0001$ indicates significance as indicated.

cavity, with an upregulation of RELMα only detectable in the IL4Rαfl/fl.CX3CR1Cre- mice (Fig 1A and 1B). Importantly, this is phenocopied in the colonic macrophages where IL4c drove strong RELMα expression in the IL4Rαfl/fl.CX3CR1Cre- but not IL4Rαfl/fl.CX3CR1Cre+ mice (Fig 1C and 1D). Likewise, in the liver, Kupffer-cell driven expression of RELMα was dependent on the absence of Cre (Fig 1E and 1F). Our results thus far therefore support the robust deletion of the IL4Rα from colonic macrophages in the IL4Rαfl/fl.CX3CR1Cre mouse, with IL4Rα signalling required for the induction of RELMα⁺ macrophages. To corroborate these data further, IL4Rα expression on colonic macrophages was determined at d35 post infection with *T. muris*, to control for the possibility of a selective outgrowth of a small population of Cre+ macrophages with incomplete deletion of the IL4Rα. These data revealed low expression of the IL4Rα on colonic macrophages in IL4Rαfl/fl.CX3CR1Cre- mice which was absent in the IL4Rαfl/fl. CX3CR1Cre+ mice (S2A Fig). Generation of bone marrow macrophages from IL4Rαfl/fl. CX3CR1Cre mice allowed a more comprehensive profiling of these macrophages and confirmed that macrophages generated from IL4Rαfl/fl.CX3CR1Cre+ mice were unable to express the

IL4Rα even following stimulation with the strong IL4Rα driver IL6 [32] (S2B Fig). Additionally, bone marrow macrophages from IL4Rαfl/fl.CX3CR1Cre+ mice were unable to upregulate the canonical alternatively activated macrophage markers CD206, RELMα and PD-L2 following exposure to IL4 (S2C–S2K Fig).

## IL4Rαfl/fl.CX3CR1Cre mice respond to *T. muris* infection with normal kinetics

Given that the IL4Rαfl/fl.CX3CR1Cre mouse is a novel transgenic mouse we characterised the expulsion kinetics and immune response to *T. muris* post-infection. We saw no differences between the worm burdens of IL4Rαfl/fl.CX3CR1Cre+ and IL4Rαfl/fl.CX3CR1Cre- mice (Fig 2A) or the quality of the immune response. Thus, as is typical of mice on a C57BL/6 background [6], infection drove a mixed IgG1/IgG2c response (Fig 2B–2G). Furthermore, a mixed cytokine response was seen in the draining mesenteric lymph nodes (MLN), with MLN cells making elevated levels of IFNγ, IL4, IL13 and TNF compared to naïve mice (Fig 2H–2K), but with no significant differences between genotype. As expected, post infection in both genotypes we saw a significant crypt hyperplasia (Fig 2L and 2M) in the descending proximal colon of the mice, which remained elevated after worm expulsion (day 35 post infection). This was accompanied by a significant goblet cell hyperplasia at d18 post infection (Fig 2L and 2N) resolving to normal levels by day 35 (Fig 2L and 2N).

Together these data demonstrate that our subsequent analyses of *in vivo* macrophage-drivers in the IL4Rαfl/fl.CX3CR1Cre mouse are not confounded by differences in expulsion kinetics or immune response between genotypes.

## Intestinal macrophages are educated towards RELMα+ macrophages in the absence of the IL4Rα subunit

To determine the role of the IL4Rα in shaping the colonic macrophage pool during *T. muris* infection, we characterised the colonic macrophage population in naïve mice and at two key time points post infection–d18 during active inflammation and at d35 during the resolution of inflammation. Analyses of macrophages within the P1-P3 gates [22–25], as well as the recently defined Tim4-CD4-, Tim4-CD4+ and Tim4+CD4+ gates within the mature (P3) macrophage population [33], revealed equivalent population shifts in both IL4Rαfl/fl.CX3CR1Cre+ and IL4Rαfl/fl.CX3CR1Cre- mice over the time course of infection (Figs 3A–3C and S3A). Thus, as anticipated, during active inflammation, we observed a significant increase in the recruitment of monocytes into the intestine with an increase in the percentage of macrophages residing within the P1 and P2 gates, accompanied by an enrichment in the Tim4-CD4- population within P3 gate, in keeping with the monocyte-derived origin of this population [33]. Based on our IL4c experiments, we hypothesized that RELMα+ macrophages in the IL4Rαfl/fl.CX3CR1Cre+ mice would be altered. However, analysis of RELMα expression within the monocyte/macrophage waterfall during both active (d18) and resolving (d35) inflammation revealed no deficit in RELMα expression in the IL4Rα null (Cre+) macrophages (Figs 3D, 3E and S3B), with both IL4Rαfl/fl.CX3CR1Cre+ and IL4Rαfl/fl.CX3CR1Cre- mice showing significant increases in RELMα expression post infection. Moreover, macrophage RELMα expression was selectively enriched in the more mature (Tim4+) P3 macrophage subsets. Thus, enhanced RELMα expression was observed in the Tim4+CD4- and Tim4+CD4+ populations compared to the Tim4-CD4- population. However the expression of RELMα by Tim4+ macrophages was seen in both the IL4Rαfl/fl.CX3CR1Cre+ and IL4Rαfl/fl.CX3CR1Cre- mice and thus independent of the IL4Rα (Fig 3F). Immunohistochemical analyses of RELMα and CD68 co-expression within the colon revealed analogous results, with no deficiency in RELMα expression in the

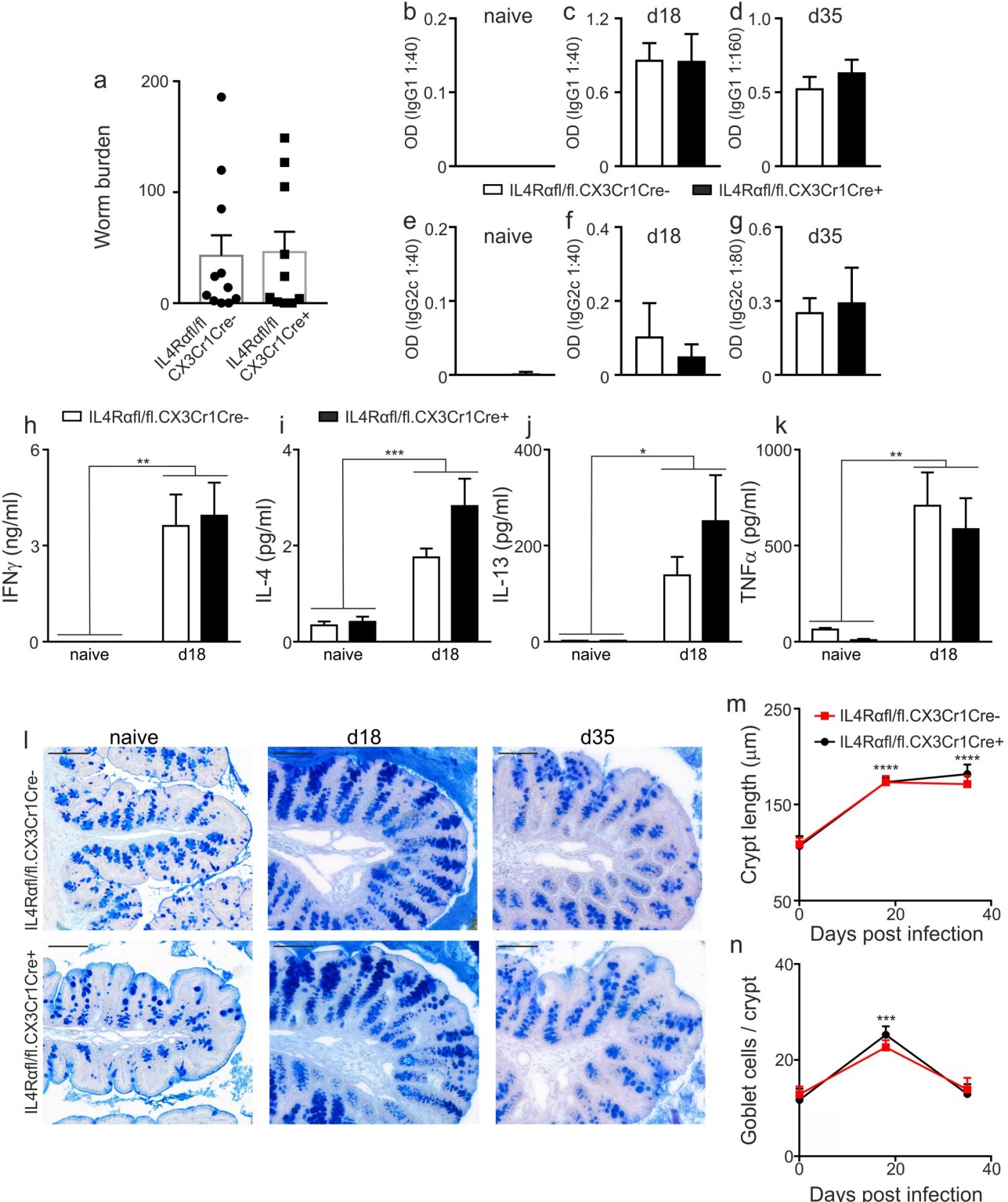

**Fig 2. The kinetics of *T. muris* infection is unaltered in the absence of IL4Rα on macrophages.** IL4Rαfl/fl.CX3CR1Cre+ or IL4Rαfl/fl.CX3CR1Cre- mice were infected with 200 *T. muris* eggs and (a) worm burden assessed in the colon and caecum at d18 post infection (n = 10–11, pooled from 2 independent experiments). Parasite-specific IgG1 from (b) naïve (n = 6–7, pooled from 2 independent experiments), (c) d18 post infection (n = 6, representative of 3 independent experiments) and (d) d35 post infection (n = 5–7, pooled from 2 independent experiments) and IgG2c production from (e) naïve, (f) d18 post infection and (g) d35 post infection. Mesenteric lymph node (MLN) cell (h) IFNγ, (i), IL4, (j) IL13 and (k) TNF profiles in

naïve mice (n = 3–4, representative of 2 independent experiments) and at d18 post infection (n = 6, representative of 3 independent experiments) cultured with *T. muris* E/S (50μg/ml) for 48h. Supernatants analysed using cytometric bead array. (l) Representative photographs of periodic acid/ Schiff's staining in colonic tissue with quantification of (m) crypt lengths and (n) goblet cell numbers in colonic tissue of naïve (n = 4–6, 2 independent experiments), d18 (n = 11–12, 2 independent experiments) and d35 (n = 5, 2 independent experiments) post infection mice. Scale bar indicates 100μm. Error bars show means ± SEM. Statistical comparisons were performed with a two-way ANOVA with post-hoc Bonferroni's multiple comparison test where appropriate: $^*p{<}0.05$, $^{**}p{<}0.01$, $^{***}p{<}0.001$, $^{****}p{<}0.0001$ indicates significance compared to naïve control.

IL4Rαfl/fl.CX3CR1Cre+ mice and no alterations in the spatial location of the RELMα⁺CD68⁺ macrophages (Fig 3G). We also analysed the expression of other signature markers associated with alternatively activated macrophages; specifically PD-L2, CD206 and Ym1, in addition to RELMα. Interestingly, although the expression of CD206 did not change post infection (Figs 3H and S3C), Ym1 (Figs 3I and S3D) was also induced *in vivo* post infection independently of IL4Rα. In contrast, induction of PD-L2 expression was IL4Rα-dependent. Thus PD-L2 expression was induced post infection in the IL4Rαfl/fl.CX3CR1Cre- mice but the % PD-L2⁺ macrophages evident in the IL4Rαfl/fl.CX3CR1Cre+ mice post infection was significantly reduced (Figs 3J and S3E). Therefore, our data reveal that the emergence of the colonic RELMα⁺Ym1⁺ macrophage is independent of IL4/IL13 during *T. muris* infection. Further, our data reveal a discord in the regulation of the alternatively activated macrophage marker PD-L2 in comparison to Ym1 and RELMα with a small IL4-dependent population of colonic macrophages present, characterised by the expression of PD-L2.

## IL4Rα deficient macrophages in a mixed bone marrow chimera express CD206, RELMα and Ym1 at levels equivalent to IL4Rα sufficient macrophages

Given our unexpected observations, we corroborated our findings using an independent experimental approach taking advantage of the IL4Rα global knockout mouse. However, as the IL4Rα global deficient mice do not develop a Th2 immune response they are unable to expel *Trichuris* and, as susceptible mice, do not drive a significant increase in RELMα⁺ macrophages in the intestine post infection [6]. Thus, RELMα macrophages only accumulate in the intestine in the *T. muris* mouse model in resistant mice. Therefore, we employed a mixed bone marrow chimera approach. Thus, CD45.1/CD45.2 recipient mice were lethally irradiated and reconstituted with a 50:50 bone marrow mix from CD45.1 (IL4Rα+/+; WT) mice and CD45.2 IL4Rα global deficient mice (IL4Rα-/-) enabling IL4Rα-dependency to be determined through analysis of CD45.1 expression (Fig 4A). At 8 weeks after irradiation mice were tail bled to analyse baseline chimerism, and mice were subsequently infected with *T. muris*. Analysis of blood chimerism revealed that blood-borne monocytes and other populations (CD4⁺ T cells, granulocytes) exhibited roughly equal proportions of CD45.1 and CD45.2 cells after reconstitution and these proportions were not altered by *T. muris* infection (S4A Fig). This is in keeping with previously published data showing IL4Rα signalling is not required for steady state survival, or survival of blood monocytes following helminth infection [31,34]. Additional analysis post-infection in both the spleen (S4B Fig) and MLNs (S4E Fig) revealed no preference for CD45.1 or CD45.2 cells. Re-stimulation of MLNs with PMA and Ionomycin following infection demonstrated a significant induction of CD4 T cell-derived IFNγ and a trend for elevated CD4 T cell-derived IL13 in infected mice compared to naïve controls (S4C and S4D Fig). In contrast, there was no upregulation of either IFNγ or IL13 derived from CD8+ T cells. Analysis of the contribution of CD45.1 and CD45.2 cells to both the whole CD4⁺ and CD8⁺ populations as well as the CD4⁺IFNγ⁺ and CD4⁺IL13⁺ populations revealed that the CD4+ cells were more likely to be from the CD45.2 (IL4Rα deficient) source. Therefore, despite lacking the IL4Rα,

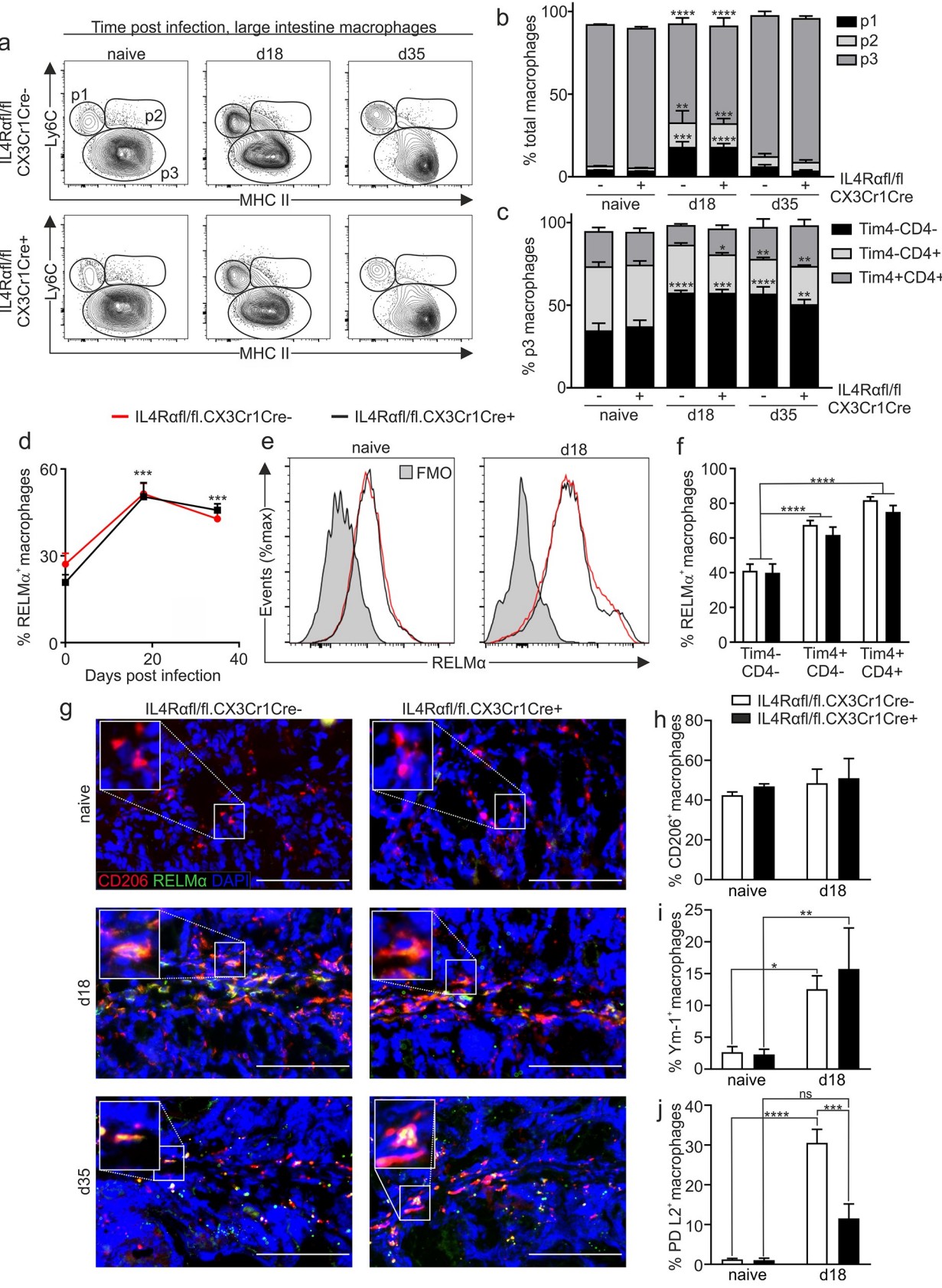

**Fig 3. Intestinal RELMα⁺ macrophages develop independently of IL4Rα expression following *T. muris* infection.** Gut macrophages (gated as CD45⁺CD11b⁺CD11c^low/int^SiglecF⁻Ly6G⁻) were divided into P1 monocyte (CD64⁻Ly6C^hi^MHCII⁻), P2 transitioning monocyte (Ly6C⁺MHCII⁺) and P3 macrophage (CD64⁺Ly6C⁻MHCII⁺) compartments in the colon and caecum. (a) Representative plots illustrating the P1-3 gates in IL4Rαfl/fl.CX3CR1Cre- and IL4Rαfl/fl.CX3CR1Cre+ mice at the defined time points. (b) Proportion of macrophages within the P1-P3 and (c) Tim4⁻CD4⁻, Tim4⁻CD4⁺ and Tim4⁺CD4⁺ gates in naïve mice and post infection with *T. muris*; naïve (n = 6–7, pooled from 2 independent experiments), d18 post infection (n = 3, representative of 3 independent experiments) and d35 post infection (n = 5–7, pooled from 2 independent experiments). Expression of the marker RELMα (d) within the total monocyte/macrophage (P1-3) population and (e) representative histograms showing RELMα expression in the IL4Rαfl/fl.CX3CR1Cre- (red) and IL4Rαfl/fl.CX3CR1Cre+ (black) naïve mice and at d18 post infection with the relevant FMO control (shaded grey) naïve (n = 6–7, pooled from 2 independent experiments), d18 post infection (n = 7–9, pooled from 2 independent experiments), d35 post infection (n = 5–7, pooled from 2 independent experiments. (f) Expression of RELMα in P3 within the Tim4⁻CD4⁻, Tim4⁻CD4⁺ and Tim4⁺CD4⁺ gates at d18 post infection (n = 7–9, pooled from 2 independent experiments). (g) Representative staining of colonic tissues stained for CD68 (Red), RELMα (Green) and counterstained with DAPI (Blue), Scale bar shows 100μm. Analysis of additional markers (h) CD206, (i) Ym1 and (j) PD-L2 in naïve mice and at d18 post infection (n = 4–7; representative of 2 independent experiments) Error bars show means ± SEM. Statistical comparisons were performed with a two-way ANOVA with post-hoc Bonferroni's multiple comparison test where applicable or by student's *t*-test. Significance is shown compared to naïve control *$p<0.05$, **$p<0.01$, ***$p<0.001$, ****$p< 0.0001$ or as indicated.

CD45.2 cells are able to produce both Th1 and Th2 cytokines (S4E Fig). This may reflect the ability of these cells to respond to other Th2-driving stimuli, for example IL33 or CCL-2 [35] and is consistent with data showing IL4 signalling is not required for the initial generation of Th2 cells in the lymph node [36]. The chimeric mice were able to respond in a typical manner to a *T. muris* infection mounting a mixed IgG1/IgG2c response (S4F and S4G Fig) and a mixed cytokine response in the draining MLNs (S4H–S4M Fig). Interestingly, and in contrast to our blood monocyte data where there was no alteration in chimerism, we saw a selection for IL4Rα-/- (CD45.2) monocytes in the colon, with a slight but significant decrease in cells derived from the CD45.1 (WT) compartment compared to in the bloodstream (Fig 4B and 4C). This occurred independently of infection. Importantly, infection with *Trichuris* drove an influx of monocytes into the colon as previously observed in the IL4Rαfl/fl.CX3CR1Cre mice with an increase in the P1 subset at day 18 post infection (Fig 4D). Analysis of the chimerism between the different monocyte and macrophage subsets in the colon again revealed a decrease in the monocytes derived from the wild type donor (CD45.1) in the intermediary P2 transiting monocyte subset in both the naïve and infected mice, however this did not appear to have a subsequent effect on the chimerism of the resident P3 macrophages (Fig 4E and 4F). Importantly, analysis of the RELMα⁺ population showed no effect of donor, demonstrating the induction of macrophage RELMα post infection with *T. muris* is entirely independent of IL4Rα expression (Fig 4G–4I), thus corroborating our data using the IL4Rαfl/fl.CX3CR1Cre mouse. Analogous results were also observed for Ym1 and CD206 expression. As in the IL4Rαfl/flCX3CR1Cre mice experiments, in our mixed bone marrow chimeras, PD-L2 expression was again dependent on the presence of IL4Rα (Fig 4H and 4I). As PD-L2 expression has been described to be exclusively observed on macrophages derived from infiltrating monocytes and not tissue resident tissue macrophages within the peritoneal cavity [37] we analysed the PD-L2+ macrophages to determine where they fell within the P1, P2 and P3 gates post infection, to determine if the altered chimerism in the P2 monocyte subset was influencing the overall PD-L2 expression. Our data shows that, in contrast to macrophages within the peritoneal cavity, PD-L2+ macrophages are predominantly found in the more mature 'P3' macrophage subset within the intestine (Fig 4J). The IL4Rα⁺ independent induction of RELMα and Ym1 on colonic macrophages is in direct contrast to data published on cavity macrophages. Thus, mixed bone marrow chimeras utilising IL4Rα deficient and sufficient bone marrow have shown that pleural cavity and peritoneal cavity macrophages express RELMα and Ym1 in an IL4Rα dependent way following infection with *Litomosoides sigmodontis* and *Heligmosomoides polygrus* respectively; the authors did not report analysis of PD-L2 expression in these studies [31]. Further, in the *Litomosoides* infection model at d16 (but not d10) post infection

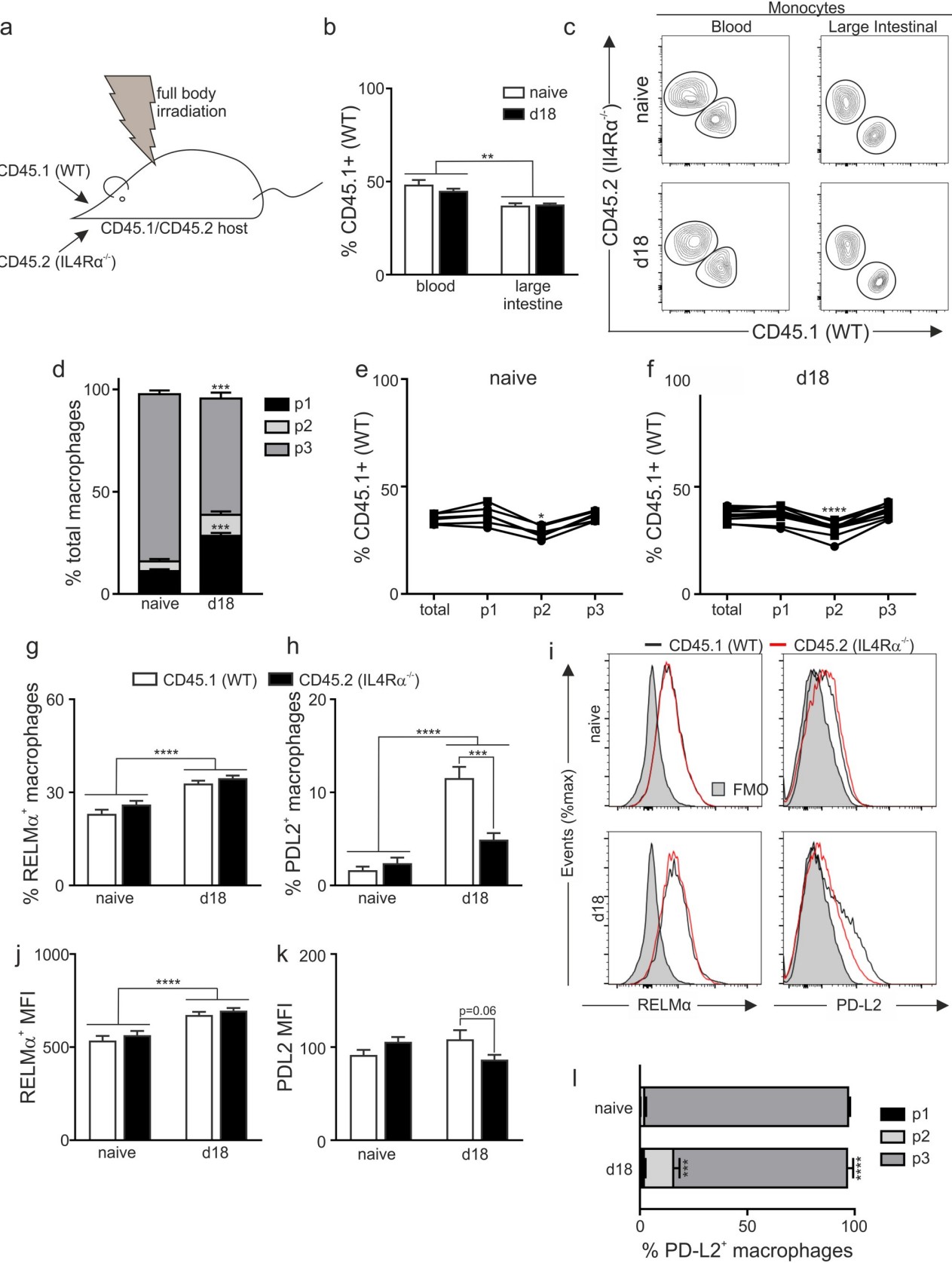

**Fig 4. IL4Rα signalling is dispensable for RELMα⁺macrophage development in the gut following *T. muris* infection of mixed bone marrow chimeras.** (a) Mixed bone marrow chimeras were generated by lethally irradiating C57BL/6 IL4Rα$^{+/+}$CD45.1$^+$CD45.2$^+$ mice and reconstitution with a 50:50 mix of IL4Rα$^{+/+}$ CD45.1 (WT) and IL4Rα$^{-/-}$ CD45.1nullCD45.2$^{+/+}$ congenic bone marrow cells; mice were left for 8 weeks to reconstitute. Mice were left uninfected or infected with 200 *T. muris* eggs and (b) Quantification of chimerism in the blood and gut monocyte population in naïve mice and at d18 post infection. (c) Representative staining for CD45.1 and CD45.2 shows chimerism of the blood monocytes (CD115$^+$) and gut monocyte (infiltrating P1 population) in naïve and infected mice. (d) Frequency of monocytes / macrophages with the P1-P3 waterfall in naïve and d18 post infection mice and the chimerism within these populations in (e) naïve and (f) infected mice. Frequency of the (g) RELMα and (h) PD-L2 expressing macrophages in the total P1-3 macrophage population from naïve and d18 post infection mice in the CD45.1 and CD45.2 populations. (i) Representative flow histograms are shown for naïve and d18 post infection mice. MFI of (j) RELMα and (k) PD-L2 expressing macrophages in the total P1-3 macrophage population from naïve and d18 post infection mice in the CD45.1 and CD45.2 populations (l) Location within the P1-P3 waterfall of PD-L2$^+$ monocytes / macrophages in naïve mice and at d18 post infection. Data are combined from two independent experiments (naïve n = 6, d18 n = 11). Error bars show means ± SEM. Statistical comparisons were performed as appropriate by repeated measure or standard two-way ANOVA with post-hoc Bonferroni's multiple comparison test where applicable. Significance is shown compared to naïve control * $p<0.05$, ** $p<0.01$, *** $p<0.001$, **** $p< 0.0001$ or as indicated.

IL4Rα expression bestowed a selective advantage specifically on the F4/80$^{hi}$ pleural macrophage population which was proposed to reflect enhanced entry of the IL4Rα$^+$ cells into the cell cycle. Therefore, the differences in IL4/IL13 dependency of macrophages seen between these studies and our data may be due in part to proliferative capacity of the macrophages analysed, with the macrophage population following *T. muris* infection not undergoing the vast proliferation seen following an infection with *Litosomoides sigmodontis*[6,38]. Thus the selective advantage provided by the presence of the IL4Rα is not observed.

Differences in IL4/IL13 dependency between gut and cavity macrophages may also reflect the length of tissue residency, with the F4/80$^{hi}$ macrophages in serous cavities which respond to helminth infections being the resident macrophage population [38], and not derived from infiltrating monocytes as observed post-infection with *T. muris*. It is possible therefore that the monocyte-derived intestinal macrophage has a less stringent dependency on IL4/IL13 signalling in order to express RELMα than resident macrophages. Indeed, using Tim4+CD4+ expression as markers of length of residency in the gut, our data showed an increased expression of RELMα within the Tim4$^-$CD4$^+$ and Tim4$^+$CD4$^+$ intestinal P3 macrophage subsets compared to the Tim4$^-$CD4$^-$ population, however this too was independent of IL4Rα expression.

## Intestinal macrophages from IL4Rαfl/fl.CX3CR1Cre mice become RELMα⁺ during *T.muris* infection even following αIFNγ treatment

The recent reports of a partial IL4Rα independence of alternatively activated macrophages in tissues other than the large intestine is particularly evident prior to the development of highly polarised type 2 adaptive immune responses [16]. Thus we sought to determine if the IL4Rα-independence of RELMα$^+$ macrophages education in the inflamed large intestine was due to the mixed Th1/Th2 environment present during an acute *T. muris* infection, as opposed to a highly polarised Th2 setting. In order to drive a highly polarised Th2 immune response, we treated IL4Rfl/fl.CX3CR1Cre mice with an αIFNγ antibody throughout the course of infection (Fig 5A). αIFNγ treatment during *T. muris* infection is a well-established methodology known to accelerate worm expulsion due to a shifting of the immune response towards a Th2 response [39,40]. Following αIFNγ treatment there was no change in the distribution of macrophages within the P1- P3 macrophage waterfall (Fig 5B and 5C). However, both parasite-specific IgG1 and IgG2c levels in the serum were significantly reduced in αIFNγ treated mice, independently of IL4Rα expression, as typically seen in mice which expel the parasite very rapidly (Fig 5D and 5E) [41]. Consistent with an enhanced Th2 response we saw a trend towards elevated serum IgE in αIFNγ depleted mice (Fig 5F). Importantly, although treatment with αIFNγ drove an increase in the percentage of RELMα$^+$ macrophages in the colon (Fig 5G) this

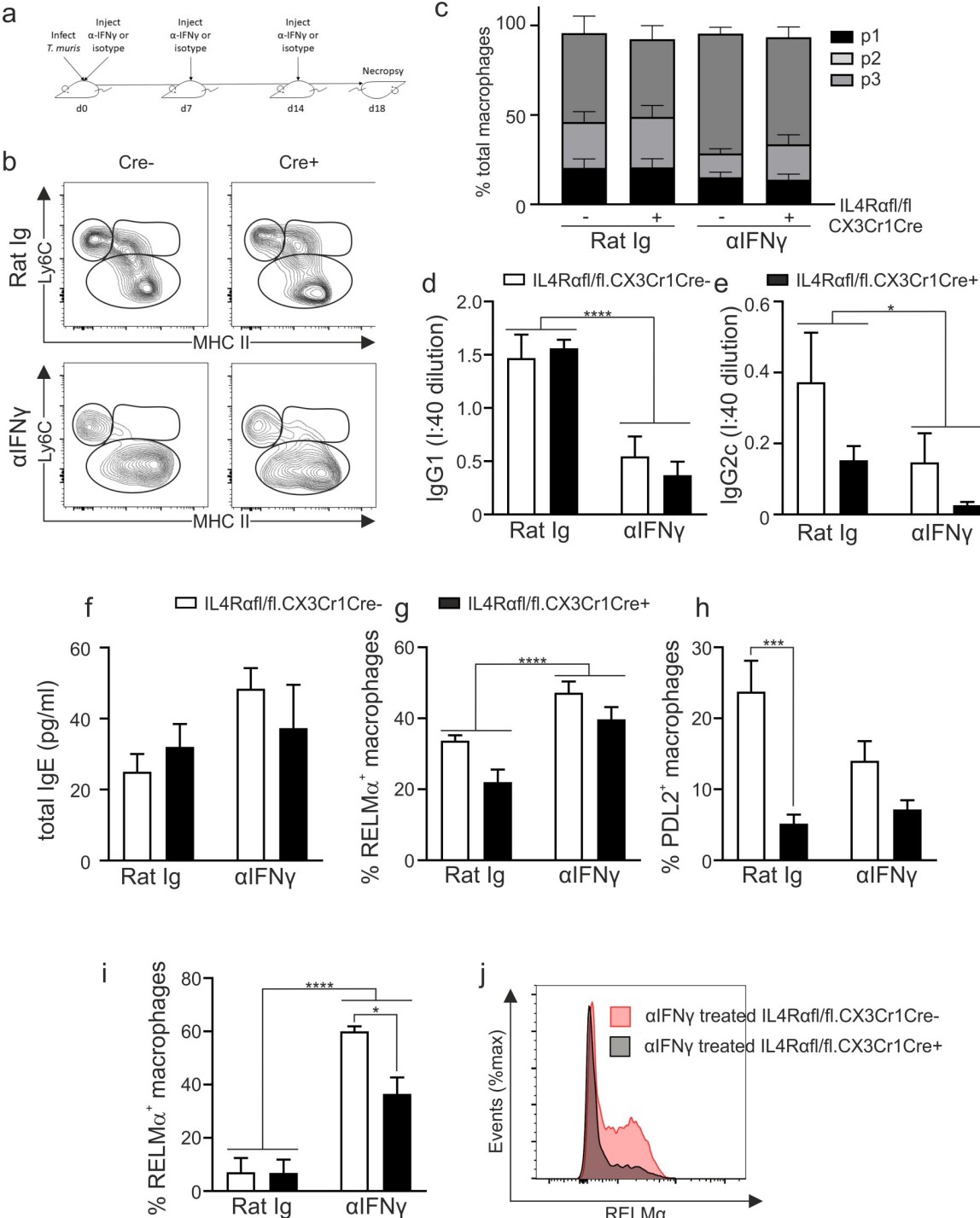

**Fig 5. Intestinal RELMα⁺ macrophages develop independently of IL4Rα expression following αIFNγ treatment during *T. muris* infection.** (a) IL4Rαfl/fl.CX3CR1Cre+ or IL4Rαfl/fl.CX3CR1Cre- mice were infected with 200 *T. muris* eggs and treated with αIFNγ or isotype mAb every 7 days during infection. Gut macrophages (gated as CD45⁺CD11b⁺CD11c^low/int^SiglecF⁻Ly6G⁻) were divided into P1 monocyte (CD64⁻Ly6C^hi^MHCII⁻), P2 transitioning monocyte (Ly6C⁺MHCII⁺) and P3 macrophage (CD64⁺Ly6C⁻MHCII⁺) compartments in the colon and caecum. (b) Representative plots illustrating the P1-3 gates in IL4Rαfl/fl.CX3CR1Cre- and IL4Rαfl/fl.CX3CR1Cre+ mice following αIFNγ or isotype mAb at d18 post infection (c) Proportion of macrophages within the P1-P3 gates post infection with *T. muris*. Parasite-specific (d) IgG1 and (e) IgG2c production and (f) total IgE production at d18 post infection. Expression of the markers RELMα (g) and PD-L2 (h) within the total monocyte/macrophage (P1-3) population. (i) Expression of RELMα within F480^hi^ MHCII^lo^ peritoneal macrophage population. (j) Representative plots illustrating the RELMα expression in the αIFNγ treated

F480$^{hi}$ MHCII$^{lo}$ peritoneal macrophages. Error bars show means ± SEM. Data are combined from two independent experiments (n = 6–7). Statistical comparisons were performed with a two-way ANOVA with post-hoc Bonferroni's multiple comparison test where appropriate: $^{*}p<0.05$, $^{**}p<0.01$, $^{***}p<0.001$, $^{****}p<0.0001$ indicates significance compared to naïve control.

occurred independently of IL4Rα expression on the macrophages. Interestingly, the PD-L2 dependence on IL4Rα expression appeared to be lost in the more strongly polarised environment suggesting that the contribution of IL4Rα signalling to PD-L2 expression may be less stringent in strong Th2 settings, although this may also reflect a return to steady state conditions due to the accelerated worm expulsion (Fig 5H).

Further, to determine if the IL4Rα independence of RELMα expression is unique to colonic macrophages we analysed the expression of RELMα on peritoneal macrophages following *T. muris* infection. In order to create a strongly polarised Th2 environment we used the same approach as used in Fig 5A for intestinal macrophages and treated groups of infected mice with either an anti-IFNγ monoclonal antibody or an isotype control. In isotype treated animals there was no significant upregulation of RELMα expression in large peritoneal macrophages post infection (Figs 1B and 5I); and no significant difference between the percentage of RELMα+ macrophages in IL4Rαfl/fl.CX3CR1Cre+ and IL4Rαfl/fl.CX3CR1Cre- mice (naïve IL4Rαfl/fl.CX3CR1Cre- 5.188±0.9823, IL4Rαfl/fl.CX3CR1Cre+ 4.477±1.632; infected isotype control treated IL4Rαfl/fl.CX3CR1Cre- 7.090±5.346 IL4Rαfl/fl.CX3CR1Cre+ 6.810±5.049; mean±SEM). However, following αIFNγ treatment, an increase in the percentage of macrophages expressing RELMα was observed in both the IL4Rαfl/fl.CX3CR1Cre- and IL4Rαfl/fl.CX3CR1Cre+ peritoneal macrophages but was significantly higher in the IL4Rαfl/fl.CX3CR1Cre- mice (Fig 5I and 5J). Therefore, whilst the expression of RELMα on colonic macrophages occurs independently of IL4Rα, expression of RELMα on large peritoneal macrophages is partially dependent on IL4Rα expression.

In conclusion, the pro- and alternatively activated properties of macrophages are linked to their activation state. Alternatively activated macrophage possess reparative functions, and thus significant interest exists in identifying how this activation state arises *in vivo* during helminth infections. Despite the physiological importance of the intestinal macrophage, a comprehensive analysis of the role of IL4Rα in driving the emergence of RELMα$^{+}$Ym1$^{+}$ macrophages in the intestine has not been shown. Indeed, it is often inferred that these macrophages are IL4Rα dependent in keeping with the main macrophage polarising signals in other tissues; but this has not been shown. Challenging prevailing paradigms, our data reveal a cell intrinsic IL4R alpha independence of the RELMα$^{+}$Ym1$^{+}$ colonic macrophage during *T. muris* infection, even in highly polarised Th2 environments. We believe this to be an important discovery given that the large intestine is a physiologically relevant compartment for many inflammatory bowel disorders. Identifying the *in vivo* signals driving this alternatively activated macrophage phenotype is thus an urgent priority.

## Methods

### Ethics statement

All experimental procedures were approved by the University of Manchester Animal Welfare and Ethical Review Board and performed within the guidelines of the Animals (Scientific Procedures) Act, 1986. Male C57BL/6 mice were purchased from Envigo at age 6–8 weeks, CX3CR1 Cre mice were a gift from S. Jung. Severe combined immunodeficient (SCID) mice, *IL4Rα$^{-/-}$* mice, *CD45.1$^{+}$CD45.2$^{+}$* mice, *CD45.1$^{+/+}$CD45.2$^{null}$* and IL4Rαfl.CX3CR1Cre mice were bred in house at the University of Manchester and used at age 6–12 weeks. Mice were

maintained at a temperature of 20–22˚C in a 12h light, 12h dark lighting schedule, in sterile, individually ventilated cages with food and water *ad lib*. All IL4Rαfl.CX3CR1Cre mice used were littermate mice. Mice were culled via exposure to increasing carbon dioxide levels, an approved schedule 1 method as specified within the guidelines of the Animals (Scientific Procedures) Act.

## *T. muris* passage

The parasite was maintained as previously described [42]. Briefly, the parasite was passaged in susceptible SCID mice through infection of these mice with 150 infective *T. muris* eggs. At day 42 post infection the caecum and colon were removed, opened longitudinally, washed in pre-warmed RPMI-1640 media (Sigma-Aldrich, UK) supplemented with penicillin (Sigma-Aldrich; 500U/ml) and streptomycin (Sigma-Aldrich; 500μg/ml) (RPMI+5xP/S). Adult *T. muris* worms were gently removed using fine forceps under a dissecting microscope and carefully inspected to ensure they retained no epithelial cells. Worms were then cultured in 6 well tissue culture plates containing RPMI + 5x P/S. Plates were incubated in a moist humidity box for 4 hours at 37˚C to collect 4h E/S then each well split into two wells containing fresh medium and incubated again in a humidity box at 37˚C overnight. Media from both the 4hr culture and the overnight culture were processed in the same way to retrieve unembryonated eggs and E/S products.

## Preparation of eggs

Media from worm cultures was centrifuged at 720g for 15 mins at room temperature. Supernatant was removed and kept for preparation of E/S. Pelleted eggs were resuspended in 40 deionised water, filtered through a 100μm nylon sieve and transferred to a cell culture flask. Flasks were examined under a dissecting microscope (Leica S8 APO) and egg density adjusted as necessary. To allow embryonation to occur, flasks were wrapped in foil to ensure eggs were kept in darkness, stored horizontally and eggs were monitored weekly. If any fungal growth was observed eggs were re-filtered or washed by centrifugation and transferred to fresh flasks. After approximately 8 weeks eggs were fully embryonated and transferred for storage, horizontally at 4˚C. In order to determine the number of eggs required to establish around 150 worms *in vivo*, all egg batches were tested in SCID mice prior to experimental use to determine the infectivity of each batch of eggs. A minimum of 3 SCID mice were infected with approximately 200 eggs and the number of larvae present in the colon and caecum at day 14 post infection determined. The infectivity was then calculated as the % larvae counted from a given number of eggs. For high-dose *T. muris* infections approximately 3ml of egg suspension was transferred to a universal tube, 15ml of deionisied water added prior to centrifugation for 15mins at 720g (low brake), room temperature. Pelleted eggs were resuspended in MQ water and embryonated eggs counted adjusted, in accordance with calculated infectivity data, to ensure there were 150 infective eggs per 200ul. Mice were then infected with 150 infective T. muris eggs in 200ul by oral gavage. During both egg counting and infection eggs were prevented from settling by stirring on a magnetic stirrer.

## Preparation of excretory/secretory products

4hr and overnight supernatants containing E/S products from passage worm cultures were filter sterilised through a 0.2μm syringe filter (Merck, Germany). E/S was then concentrated using Amicon Ultra-15 centrifugal filter units (Merck) by centrifugation at 3,000g at 4˚C. Centrifugation was carried out until approximately 20% of the original volume of supernatant remained. E/S was then dialysed against PBS using Slide-A-Lyzer Dialysis cassettes, 3.500

MWCO (Thermo Scientific, UK) at 4˚C. The E/S was then filtered sterilised a final time through a 0.2μm syringe filter, the protein concentration measured and aliquoted prior to storage at -80˚C

## Quantification of *T. muris* worms

At necropsy the caecum and proximal colon were collected, blinded and stored at -20˚C. For worm counts, frozen caecum and colon were defrosted in a petri dish containing water and cut open longitudinally. The gut contents were removed through swilling in the water and the gut tissue transferred to a fresh petri dish containing water. The gut mucosa was scraped off using curved forceps to remove epithelia and worms from the gut tissue. Both the gut contents and removed gut mucosa were then examined and *T. muris* worms counted using a dissecting microscope (Leica S8 APO).

**In vivo antibody treatment.** Monoclonal antibodies against IFNγ (XMG-1.2) and control isotype (GL113) were a gift from L. Boon. Mice were treated intraperitoneally with 1 mg of antibody on day 0, day 7 and day 14 post infection.

**In vivo IL4 complex treatment.** IL-4–anti–IL-4 mAb complex (IL4c) was generated by combining 5 μg of recombinant IL-4 (PeproTech) with 25 μg 11B11 (Bio X Cell) per mouse. Mice were treated intraperitoneally with IL4c and tissue harvested 48 hours later.

## Bone marrow chimeras

Competitive mixed BM chimeric mice were created by lethally irradiating C57BL/6 *CD45.1$^+$CD45.2$^+$* mice with 11.5 Gy γ radiation administered in two doses ∼3 h apart, followed by i.v. injection of $9 \times 10^6$ BM cells depleted of mature T cells using CD90 microbeads (Miltenyi Biotec) and comprised of a 1:1 mix of cells from C57BL/6 *CD45.2$^{+/+}$IL4Rα$^{-/-}$* mice and C57BL/6 *CD45.1$^{+/+}$CD45.2$^{null}$* mice. Chimeric animals were left 8 weeks before further experimental manipulation.

## Tissue preparation and cell isolation

**Caecum and colon lamina propria.** Cells were isolated as previously described with some modifications [43]. In brief, after dissection of the caecum and colon, the colonic patch was removed from the tip of the caecum and all adherent adipose tissue removed from the tissue. The caecum and colon were cut longitudinally and washed thoroughly with pre-chilled PBS on ice. To remove intestinal epithelial cells and leukocytes, the tissue was incubated in pre-warmed media (RPMI 1640) supplemented with 3% FCS (Sigma-Aldrich), 20 mM Hepes (Sigma-Aldrich), 5 mM EDTA (VWR, UK), and 1 mM freshly thawed dithiothreitol (Sigma-Aldrich) for 20 min at 37˚C with agitation. After incubation, gut segments were shaken three times in fresh prewarmed RPMI 1640 (serum-free) with 2 mM EDTA and 20 mM Hepes. The remaining tissue was minced with fine scissors and digested at 37˚C for 30 min with continuous stirring at 450 rpm in serum-free RPMI 1640 containing 20 mM Hepes, 0.1 mg/ml liberase TL (Sigma-Aldrich), and 0.5 mg/ml DNase I (Sigma-Aldrich). Digested tissue was passed sequentially through a 70-μm filter and 40-μm cell strainer, and after pelleting, it was resuspended in media supplemented with 10% FCS until staining.

**Blood.** Blood was collected into heparin-coated 0.2mm capillary tubes (VWR) and stored in EDTA (VWR). Red blood cells were lysed in RBC lysis buffer (Sigma-Aldrich) for 5 min on ice twice. Cells were then washed and resuspended in PBS containing 10% FCS until staining.

**Culture of bone marrow macrophage cells.** Bone marrow-derived macrophages (BMDM) were isolated from mice by flushing femurs with Dulbecco's modified Eagle's medium (DMEM; Invitrogen) containing 10% FCS, 1% L-glutamine and 100U/ml penicillin/

100μg/ml streptomycin (all from Sigma-Aldrich, UK) (complete DMEM). Cells were washed and cultured at $1 \times 10^6$/ml in complete DMEM containing 30 ng/ml M-CSF (Peprotech, UK) for 7 days, with the media being replaced after 4–5 days. Macrophage purity was assessed by flow cytometry and in all cultures used the purity was found to be over 90%. The BMDMs $(0.5 \times 10^6$/ml) were stimulated for 24 hours with IL4 (20ng/ml; Peprotech) or for 18 hours with IL6 (50ng/ml; Peprotech). Cells were removed using Accutase (Thermo Scientific) prior to flow analysis.

## Flow cytometry

Single-cell suspensions of caecum/colon, blood, bone marrow derived macrophages, MLN or spleen were washed thoroughly with PBS and stained with the Live/Dead Fixable blue dead cell stain kit (Thermo Scientific) in the dark for 15 min at 4˚C to exclude dead cells. Subsequently, cells were stained in the dark for 10 min at 4˚C with anti-CD16/CD32 (eBioscience) in PBS containing 0.5% BSA (Sigma-Aldrich) and a further 20min with the relevant fluorochrome-conjugated antibodies (see Table 1) in Brilliant stain buffer (BD biosciences, UK).

**Table 1. Flow cytometry antibodies used.**

| | | | |
|---|---|---|---|
| CD4 | GK1.5 | FITC or BV711 | Biolegend |
| CD11b | M1/70 | FITC or BV711 | Biolegend |
| CD11c | N418 | BV605 | Biolegend |
| CD45 | 30F11 | AF700 | Biolegend |
| CD45.1 | A20 | AF700 | Biolegend |
| CD45.2 | 104 | FITC | Biolegend |
| CD64 | X54-5/7.1 | BV421 | Biolegend |
| CD115 | AFS98 | APC or PE-Cy7 | Biolegend |
| MHCII (I-A/I/E) | M5/114.15.2 | PE-Dazzle594 or BV650 | Biolegend |
| Tim-4 | RMT4-54 | PE-Cy7 | Biolegend |
| Ly6C | HK1.4 | APC/Fire750 | Biolegend |
| Siglec F | E50-2440 | PerCP-Cy5.5 | BD biosciences |
| TCRβ | H57-597 | PerCP-Cy5.5 | Biolegend |
| CD8α | 53–6.7 | BV605 | Biolegend |
| IFNγ | XMG1.2 | PE-Cy7 | Biolegend |
| IFNγ iso | RTK2071 | PE-Cy7 | Biolegend |
| IL13 | eBio13A | PE | eBiosciences |
| IL13 iso | eBRG1 | PE | eBiosciences |
| Ki67 | SolA15 | eF450 | eBiosciences |
| Ki67 | 16A8 | APC | Biolegend |
| CD44 | 1M7 | PE or APC-Cy7 | Biolegend |
| RELMα | - | Purified | Peprotech |
| iNOS | CXNFT | PE | eBiosciences |
| PD-L2 | TY25 | PE | Biolegend |
| Ym1 | - | Biotin | R&D Systems |
| CD206 | C068C2 | PE-Cy7 | Biolegend |
| Ly6G | 1A8 | PerCP-Cy5.5 | Biolegend |
| CD124 (IL4Rα) | mIL4R-M1 | PE | BD biosciences |
| F4/80 | BM8 | PE-Cy7, PE or FITC | Biolegend |
| F(ab')2-Gt anti-Rb IgG (H+L) | - | APC | Thermo Scientific |
| SA-BV711 | - | - | Biolegend |
| SA-BV605 | - | - | Biolegend |

Cells were washed and in some cases, cells were immediately acquired live, or alternatively, cells were fixed in fixation buffer (True-Nuclear Transcription factor buffer set, Biolegend, UK) for 20 min at 4˚C and resuspended in PBS containing 0.5% BSA. For intracellular staining cells were permeabilised (True-Nuclear Transcription factor buffer set, Biolegend) and stained in permeabilisation buffer with the relevant fluorochrome-, biotin- conjugated or purified antibodies for 20 min at 4˚C. Cells were washed in permeabilisation buffer and resuspended when relevant with fluorochrome-conjugated secondary antibodies and then washed in PBS prior to resuspending in PBS containing 0.5% BSA for acquisition. Cell acquisition was performed on an LSR Fortessa or LSR II running FACSDIVA 8 software (BD biosciences). For each intestinal sample, typically 10,000–20,000 macrophages were collected. Data were analysed using FlowJo software version 10 (TreeStar, UK). For gating strategies see S5–S8 Figs.

### Gut monocyte and macrophage isolation

Single-cell suspensions were prepared as described above with the following modifications: incubation with dithiothreitol and EDTA was reduced to 10 min, and the liberase digestion step was decreased to 20 min but with an increased concentration of liberase TL (0.75 mg/ml). Cells were sorted on an Influx (BD biosciences) and isolated cells were suspended in PBS supplemented with 2% FCS (Sigma-Aldrich) and 2 mM EDTA (VWR). Sorted cells were collected in PBS with 10% FCS and stored on ice prior to washing and resuspending in Trizol (Thermo Fisher) before storage at −80˚C for subsequent RNA extraction.

### RNA extraction

RNA was extracted from sorted cells using an RNeasy mini kit (QIAGEN, UK) following the manufacturer's instructions.

### RNA sequencing and analysis

Total RNA was submitted to the University of Manchester Genomic Technologies Core Facility. Quality and integrity of the RNA samples were assessed using a 2200 TapeStation (Agilent Technologies) and then libraries generated using the TruSeq Stranded mRNA assay (Illumina, Inc.) according to the manufacturer's protocol. Briefly, total RNA was used as input material from which polyadenylated mRNA was purified using poly-T, oligo-attached, magnetic beads. The mRNA was then fragmented using divalent cations under elevated temperature and then reverse transcribed into first strand cDNA using random primers. Second strand cDNA was then synthesised using DNA Polymerase I and RNase H. Following a single 'A' base addition, adapters were ligated to the cDNA fragments, and the products then purified and enriched by PCR to create the final cDNA library. Adapter indices were used to multiplex libraries, which were pooled prior to cluster generation using a cBot instrument. The loaded flow-cell was then paired-end sequenced (76 + 76 cycles, plus indices) on an Illumina HiSeq4000 instrument. Finally, the output data was demultiplexed (allowing one mismatch) and BCL-to-Fastq conversion performed using Illumina's bcl2fastq software, version 2.17.1.14

Unmapped paired-end sequences from an Illumina HiSeq4000 sequencer were tested by FastQC (http://www.bioinformatics.babraham.ac.uk/projects/fastqc/). Sequence adapters were removed and reads were quality trimmed using Trimmomatic_0.36 (PMID: 24695404). The reads were mapped against the reference mouse genome (mm10/GRCm38) and counts per gene were calculated using annotation from GENCODE M14 (http://www.gencodegenes.org/) using STAR_2.5.3 (PMID: 23104886). Normalisation, Principal Components Analysis, and differential expression was calculated with DESeq2_1.16.1 (PMID:25516281). RNA sequencing data has been uploaded to ArrayExpress, reference: E-MTAB-10501.

## Mesenteric lymph node (MLN) cell re-stimulation and cytokine bead array

MLN cells were brought to cell suspension and $5\times10^6$ cells/ml were cultured for 48h at 37˚C 5% $CO_2$ in RPMI 1640 with 4 hr E/S antigen (50μg/ml). Supernatants were harvested and stored at −20˚C until they were assayed for cytokines. Cytokines were analysed using the Cytometric Bead Array (CBA) Mouse/Rat soluble protein flex set system (BD Bioscience), which was used according to the manufacturer's instructions. Cell acquisition was performed on a FACS Verse (BD Biosciences) or MACSQuant (Miltenyi Biotech). For analysis, FCAP Array v3.0.1 software (BD Cytometric Bead Array) was used. For intracellular cytokine analysis MLN cells were brought to cell suspension and $5\times10^6$ cells/ml were cultured for 16h at 37˚C 5% $CO_2$ with eBioscience Cell Stimulation cocktail (Thermo Fisher Scientific) prior to flow cytometric analysis.

## Histology

Proximal colon tissue was fixed for 24 hours in 10% neutral buffered formalin (Fisher) containing 0.9% sodium chloride (Sigma-Aldrich), 2% glacial acetic acid (Sigma-Aldrich) and 0.05% alkyltrimethyl-ammonium bromide (Sigma-Aldrich) prior to storage in 70% Ethanol (Fisher) until processing. Fixed tissues were dehydrated through a graded series of ethanol, cleared in xylol and infiltrated with paraffin in a dehydration automat (Leica ASP300 S) using a standard protocol. Specimens were embedded in paraffin (Histocentre2, Shandon), sectioned on a microtome (5μm sections) and allowed to dry for a minimum of 4 hours at 40˚C. Prior to staining slides were deparaffinised with citroclear (TCS biosciences) and rehydrated through alcohol (100% to 70%) to PBS or water.

Mucins in goblet cells were stained with 1% alcian blue (Sigma-Aldrich) in 3% acetic acid (Sigma-Aldrich, pH 2.5) for 5 mins. Sections were washed and treated with 1% periodic acid, 5mins (Sigma-Aldrich). Following washing sections were treated with Schiff's reagent (Vicker's Laboratories) for 15mins and counterstained in Mayer's haematoxylin (Sigma-Aldrich). Slides were then dehydrated and mounted in depex mounting medium (BDH Laboratory Supplies). For enumeration of goblet cell staining, the average number of cells from 20 crypts was taken from three different sections per mouse. Images were acquired on a 3D-Histech Pannoramic-250 microscope slide-scanner using a *20x/ 0.30 Plan Achromat* objective (Zeiss).

## Immunohistochemistry

Proximal colon tissue was embedded in OCT and snap-frozen on dry ice and stored at -80C until processing. Cryostat frozen sections (6 μm) were fixed in acetone, blocked with blocking solution (Perkin-Elmer) plus 7% goat serum (Vector) prior to incubation with 2 μg/mL rabbit anti-mouse RELMa (Peprotech) followed by anti-rabbit Alexa488 (2.5 μg/ml, Invitrogen). Sections were re-blocked with blocking solution (Perkin-Elmer) plus 7% rat serum (Vector) prior to incubation with rat anti-mouse CD68-AF647 (1 μg/ml, BioLegend). Sections were counterstained and mounted with VectaShield Hard set mounting media with DAPI (Vector). Images were collected on a *Zeiss Axioimager D2* upright microscope using a *20x / 0.5 EC* Plan Apochromat objective and captured using a Coolsnap HQ2 camera (Photometrics) through Micromanager software v1.4.23. Specific band pass filter sets for *DAPI, FITC and Cy5* were used to prevent bleed through from one channel to the next. Images were then processed and analysed using *Fiji ImageJ (*http://imagej.net/Fiji/Downloads*)*.

## IgG ELISA

Serum was assayed for parasite specific IgG1 and IgG2c antibody production. 96 well plates were coated with 5μg/ml *T. muris* overnight E/S antigen overnight, plates were washed, and

non-specific binding blocked with 3% BSA (Sigma-Aldrich) in PBS. Following washing, plates were incubated with serum (2 fold dilutions, 1:20–1:2560) and parasite specific antibody was measured using biotinylated IgG1 (BD Biosciences) or IgG2c (BD Biosciences) antibodies which were detected with SA-POD (Roche). Finally, plates were washed and developed with TMB substrate kit (BD Biosciences, Oxford, UK) according to the manufacturer's instructions. The reaction was stopped using 0.18 M $H_2SO_4$, when sufficient colour had developed. The plates were read by a Versa max microplate reader (Molecular Devices) through SoftMax Pro 6.4.2. software at 450 nm, with reference of 570 nm subtracted.

## IgE ELISA

Serum was assayed for total IgE antibody production. 96 well plates were coated with purified anti-mouse IgE (2ug/ml, Biolegend, Clone: RME-1) in 0.05M carbonate/ bicarbonate buffer and incubated overnight at 4˚C. Following coating, plates were washed in PBS-Tw and non-specific binding blocked with 3% BSA (Sigma-Aldrich) in PBS for 1 hour at room temperature. Plates were washed and diluted serum (1:10) added to the plate and incubated for 2hrs at 37˚C. After washing HRP conjugated goat anti-mouse IgE (1ug/ml; Bio-rad) was added to the plates for 1 hour. Finally, plates were washed and developed with TMB substrate kit (BD Biosciences, Oxford, UK) according to the manufacturer's instructions. The reaction was stopped using 0.18M $H_2SO_4$, when sufficient colour had developed. The plates were read by a Versa max microplate reader (Molecular Devices) through SoftMax Pro 6.4.2. software at 450 nm, with reference of 570 nm subtracted.

## Statistical analysis

All raw data is available (S1 Data). Comparisons between groups were undertaken using Prism (8.0; GraphPad Software). Two experimental groups were compared using a Student's *t* test, where more than two groups were compared, a one-way ANOVA or two-way ANOVA was used as appropriate. Significance was set at $p \leq 0.05$.

## Supporting information

**S1 Fig. Confirmation of experimental models. Alternatively activated macrophages accumulate following T. muris infection.** Gut macrophages (gated as CD45+CD11b+CD11clow/intSiglecF-Ly6G-) were FACS-sorted from infected C57 BL6/J mice at d35 post infection based on expression of CD206 and RNA-Seq performed. (a) CD206 was chosen as a good cell surface surrogate for RELMα (b) quality control metrics, summary of reads counting. (c) PCA analysis of RNAseq data and (d) volcano plot analysis highlighting genes of interest. (TIF)

**S2 Fig. IL4Rαfl/fl.CX3CR1Cre+ macrophages are unresponsive to IL4 signalling.** (a) IL4Rα expression was determined by flow cytometric analysis in gut macrophages at d35 post *T. muris* infection (n = 3) or (b) Bone marrow derived macrophages (BMDMs) from IL4Rαfl/fl. CX3CR1Cre+ and IL4Rαfl/fl.CX3CR1Cre- mice cultured for 18 hours in the presence or absence of IL6 for 18 hours. BMDMs were cultured in the presence of absence of IL4 for 24 hours and the expression of (c-e) CD206, (f-h) PD-L2 and (i-k) RELMα determined by flow cytometry. BMDM data is from 4–5 independent biological repeats. Error bars show means ± SEM. Statistical comparisons were performed with a two-way ANOVA with post-hoc Bonferroni's multiple comparison test where applicable or by student's *t*-test. Significance is shown compared to naïve control $^*p<0.05$, $^{**}p<0.01$, $^{***}p<0.001$, $^{****}p< 0.0001$ or as

indicated.
(TIF)

**S3 Fig. Intestinal RELMα⁺ macrophages develop independently of IL4Rα expression following *T. muris* infection.** (a) Representative plots illustrating the Tim4⁻CD4⁻, Tim4⁻CD4⁺ and Tim4⁺CD4⁺ gates in IL4Rαfl/fl.CX3CR1Cre- and IL4Rαfl/fl.CX3CR1Cre+ mice at the defined time points. (b) MFI of the marker RELMα within the total monocyte/macrophage (P1-3) population in naïve (n = 6–7, pooled from 2 independent experiments), d18 post infection (n = 7–9, pooled from 2 independent experiments) and d35 post infection (n = 5–7, pooled from 2 independent experiments) mice. MFI analysis of additional markers (c) CD206, (d) Ym1 and (e) PD-L2 at d18 post infection (n = 4–7; representative of 2 independent experiments). Error bars show means ± SEM. Statistical comparisons were performed with a two-way ANOVA with post-hoc Bonferroni's multiple comparison test where applicable or by student's *t*-test. Significance is shown compared to naïve control $^*p<0.05$, $^{**}p<0.01$, $^{***}p<0.001$, $^{****}p<0.0001$ or as indicated.
(TIF)

**S4 Fig. Mixed bone marrow chimeras show no survival advantage of IL4Rα expressing macrophages and mount a mixed Th1/Th2 response to *T. muris* infection.** Mixed bone marrow chimeras were generated by lethally irradiating C57BL/6 IL4Rα⁺/⁺CD45.1⁺CD45.2⁺ mice and reconstitution with a 50:50 mix of IL4Rα⁺/⁺ CD45.1 and IL4Rα⁻/⁻ CD45.1ⁿᵘˡˡCD45.2⁺/⁺ congenic bone marrow cells; mice were left for 8 weeks to reconstitute. (a) Mice were tail bled prior to being left uninfected or infected with 200 *T. muris* eggs and blood chimerism analysed by assessing frequency of the CD45.1⁺ (WT) cells at d0 and d18 post infection, with lines joining individual mice. (b) Spleen populations analysed in naïve mice and at 18 post infection, with lines joining individual mice. CD4⁺ and CD8⁺ T cells from the draining mesenteric lymph nodes were re-stimulated with eBioscience cell stimulation cocktail and intracellular cytokine expression of (c) IFNg and (d) IL13 analysed. (e) Frequency of the IL4Rα⁺/⁺CD45.1 cells within the different T cell populations was analysed at d18 post infection, with lines joining cells of individual mice. Parasite-specific (f) IgG1 and (g) IgG2c production in blood serum from naïve mice and at d18 post infection. MLN (h) IFNγ, (i) IL10, (j) TNF, (k) IL17A, (l) IL6 and (m) IL13 profiles in naïve mice and at d18 post infection cultured with *T. muris* E/S (50μg/ml) for 48h. Data are combined from two independent experiments (naïve n = 6, d18 n = 11). Error bars show means ± SEM. Statistical comparisons were performed as appropriate by repeated measure or standard two-way ANOVA with post-hoc Bonferroni's multiple comparison test where applicable. Significance is shown compared to naïve control $^*p<0.05$, $^{**}p<0.01$, $^{***}p<0.001$, $^{****}p<0.0001$ or as indicated.
(TIF)

**S5 Fig. Peritoneal cavity gating strategy.**
(TIF)

**S6 Fig. Colon gating strategy.**
(TIF)

**S7 Fig. Liver gating strategy.**
(TIF)

**S8 Fig. Bone marrow chimera colon gating strategy.**
(TIF)

**S1 Data. Raw data for all main and supplementary figures.**
(XLSX)

## Acknowledgments

Thanks goes to the following core facilities at The University of Manchester: the bioimaging, histology, flow cytometry, genomic technologies, bioinformatics, and biological services facilities. Irradiation in these experiments was performed with assistance from Epistem Ltd and Dr Joanne Konkel. Special thank goes to Roger Meadows, Steve Marsden, Gareth Howell, Matthew Brown, Leo Zeef and Andy Hayes for their technical assistance. We thank Dr Steve Jenkins, Dr Tara Sutherland, Dr John Grainger and Dr Dominik Ruckerl for their discussions of the data and experimental design.

## Author Contributions

**Conceptualization:** Ruth Forman, Judith E. Allen, Werner Muller, Joanne L. Pennock, Kathryn J. Else.

**Data curation:** Ruth Forman, Hannah Smith, Kelly Wemyss, Iris Mair.

**Formal analysis:** Ruth Forman.

**Funding acquisition:** Ruth Forman, Judith E. Allen, Werner Muller, Joanne L. Pennock, Kathryn J. Else.

**Investigation:** Ruth Forman, Larisa Logunova, Hannah Smith, Kelly Wemyss, Iris Mair, Kathryn J. Else.

**Methodology:** Ruth Forman, Larisa Logunova, Hannah Smith, Kelly Wemyss, Iris Mair.

**Project administration:** Ruth Forman, Kathryn J. Else.

**Resources:** Louis Boon, Judith E. Allen, Kathryn J. Else.

**Supervision:** Werner Muller, Joanne L. Pennock, Kathryn J. Else.

**Visualization:** Ruth Forman, Kathryn J. Else.

**Writing – original draft:** Ruth Forman, Kathryn J. Else.

**Writing – review & editing:** Ruth Forman, Hannah Smith, Kelly Wemyss, Iris Mair, Judith E. Allen, Werner Muller, Joanne L. Pennock, Kathryn J. Else.

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
