## [Decision Letter · Decision Letter 0]

25 Mar 2021

Dear Dr Forman,

Thank you very much for submitting your manuscript "Trichuris muris infection drives cell-intrinsic IL4R alpha independent large intestinal RELMα+ macrophages." for consideration at PLOS Pathogens. As with all papers reviewed by the journal, your manuscript was reviewed by members of the editorial board and by several independent reviewers. In light of the reviews (below this email), we would like to invite the resubmission of a significantly-revised version that takes into account the reviewers' comments.

All reviewers agree on the importance of this study, which will be of interest to the mucosal research community, however, suggest important experiments to validate the conclusions, as well as text changes so as not to overinterpret the findings. Please address all reviewers comments.

We cannot make any decision about publication until we have seen the revised manuscript and your response to the reviewers' comments. Your revised manuscript is also likely to be sent to reviewers for further evaluation.

Sincerely,

Meera Goh Nair

Associate Editor

PLOS Pathogens

P'ng Loke

Section Editor

PLOS Pathogens

Kasturi Haldar

Editor-in-Chief

PLOS Pathogens

orcid.org/0000-0001-5065-158X

Michael Malim

Editor-in-Chief

PLOS Pathogens

orcid.org/0000-0002-7699-2064

All reviewers agree on the importance of this study, which will be of interest to the mucosal research community, however, suggest important experiments to validate the conclusions, as well as text changes so as not to overinterpret the findings. Please address all reviewers comments.

Reviewer's Responses to Questions

**Part I - Summary**

Reviewer #1: The manuscript by Forman et al. entitled "Trichuris muris infection drives cell-intrinsic IL4R-alpha independent large intestinal RELMα+ macrophages" addresses an important and relevant gap of knowledge by investigating the role of IL-4Ralpha responsiveness of macrophages present in the large intestine during the helminth Trichuris muris infection.

Although the results report an effective impairment of the IL4Ra expression in macrophages, there was no significant effect of macrophage IL4Ra on worm burden, antibody responses, cytokine response in the draining lymph node, gut inflammation or intestinal macrophage subset responses. In line with these observations, macrophages in the large intestine expressed more RELMa upon worm infection, but RELMa (as YM1) expression was not affected by the absence of macrophage IL4Ra (using CX3CR1creIl4rafloxed mice and WT-IL4Ra-/- mixed bone marrow chimeric mice). The authors conclude that macrophages upregulate RELMa (and YM1, but not PD-L2) independently of IL4Ra signaling which somehow explains the absence of differences in worm immunity. This is an interesting observation but the manuscript does not fully confirm the conclusions in its current form.

Reviewer #2: In this work Forman et al investigate the requirement of IL-4R alpha to drive RELM-alpha expression in large intestinal macrophages during Trichuris muris infection.

The main findings include the generation and characterisation of a mouse strain in which IL-4R alpha is missing in CX3CR1+ cells, which provides a new angle in the study of IL-4R alpha-mediated signalling in vivo as this promoter drives efficient Cre expression in all intestinal macrophages. According to the authors the LySM Cre driver does not induce efficient deletion of floxed genes in the gut, particularly during inflammation. This new tool will be of interest to the researchers investigating the role of M(IL-4) macrophages in vivo.

The authors then show that IL-4R alpha deficiency in macrophages does not alter the kinetics of T. muris infection and investigate the phenotype of macrophage populations at d18 and d35 post-infection. They find no differences in RELM-alpha expression in the absence of IL-4R-alpha but reduced PDL2 after infection.

The authors then confirmed their observations using a mixed bone marrow chimeras of WT and global IL-4Ralpha deficient mice showing that in the presence of IL-4R alpha competent macrophages, still the IL-4Ralpha negative macrophages expressed RELM-alpha during infection but did not upregulate PDL2.

Finally, the authors investigated if polarisation of mice towards the Th2 phenotype using a blocking antibody against IFN-gamma alters their findings. This model has the caveat of a faster worm expulsion so data might need to be interpreted with caution. This is highlighted by the authors. In this setting REM-alpha expression in macrophages remains independent of IL-4R-alpha signalling. PDL-2 expression is reduced and becomes less dependent on IL-4R alpha expression.

Reviewer #3: In this study, Forman et al address the question of whether colonic macrophages that are induced following Trichuris muris infection and adopt an alternatively activated/M(IL4) phenotype (ie express RELMa, Ym1, CD206, PDL2) are IL4Ra-dependent. Surprisingly, despite a transcriptional signature that is consistent with IL-4-elicited macrophage polarization, up-regulation of RELMa and Ym1 in colonic macrophages is independent of IL-4Ra in a cell intrinsic manner at two different time points following T muris infection. Authors use two independent genetic approaches to address the IL4Ra-independent phenotype, including a the CX3CR1 macrophage specific Cre driver. The data they show supports their conclusion that T muris induced colonic macrophages express RELMa-independently of IL4Ra (as stated in the title), but language used throughout the text attempts to enhance the application of these findings more broadly (for example, in relation to IBD, colonic education of macrophages). Authors are encouraged to make clear distinctions between claims their data can support and avenues of interest that remain speculative. Otherwise, the study includes relevant controls and describes data of interest in the field of mucosal immunology.

**Part II – Major Issues: Key Experiments Required for Acceptance**

Reviewer #1: Major comments:

1. The absence of IL4Ra expression in macrophages is not fully characterized in colonic macrophages. The results are in Fig 1c and d are however convincing as IL-4c treatment did not results in the detection of an increased proportion of RELMa+ macrophages. However, there is no characterization of the CX3CR1cre Il4ra-floxed mouse. As acknowledged by the authors, CX3CXR1 is not restricted to macrophages and other cell types in these mice will also have impaired IL4Ra expression. This should be addressed in the steady-state and during infection (MFI for IL4Ra, RT-qPCR for Il4ra expression, and potentially genotyping of Cre-mediated deletion efficiency of the targeted exons) in sorted populations of macrophages. The results section refers to Yona et al to state that CX3CR1 is expressed in all macrophage subsets in the intestine, but this study describes the macrophage populations in homeostasis, not during inflammation.

2. The overall RNAseq analysis should be better detailed, such as PCA analysis and other QC parameters. Also, the rationale to choose the CD206+/- populations for analysis is described but is hardly followed in the following sections of the manuscript. The heatmaps that are shown showing the 20 genes up- or down-regulated only provide a limited information about the data, and also hide the statistical significance of the observed fold changes. A Volcanoplot (log2FC vs P values) and/or MA plot (normalized counts vs log2 FC) would be informative about the quality of the data and significance. Then, the conclusion that CD206+ macrophages are anti-inflammatory based on the regulated gene might well be true, but performing gene set enrichment analyses would be a substantial added value to support the conclusions.

3. The authors observed RELMa+ macrophages in CX3CR1creIl4rafloxed mice, as well as in WT:IL4Ra-/- chimeras. This is an interesting observation, although the increased RELMa expression after T. muris infection in macrophages does not seem to be very strong (between naïve and infected mice). However, the authors do not show the data of IL4Ra-/- mice, and whether RELMa (and YM1, CD206, PD-L2) is also up in these mice. In contrast to PD-L2, RELMa and YM1 are secreted proteins that could be well produced by other another cell type in large quantities in response to IL-4 and IL-13 (or not) and phagocytosed by intestinal macrophages. This phenomenon was found in alveolar macrophages after N. brasiliensis infection (Krljanac et al., 2019 – ref 14).

Before concluding that intestinal macrophages express RELMa at similar levels in both mouse strains, RNA expression should be determined specifically in these cells to determine the expression levels of the Retnla gene. Unfortunately, the immunostaining in colon tissue sections does not allow to draw a clear conclusion.

4. The choice of CX3CR1Cre instead of LysMCre is explained in the results-discussion section, with the given explanation that Lyz2 is not sufficiently expressed in gut macrophages during Schistosoma mansoni infection (Vanella et al, 2014). However, gut macrophages were not addressed in this study and it would have been relevant to compare the CX3CR1cre IL4rafloxed mice with LysMCre IL4rafloxed.

Reviewer #2: In view of these findings I wonder if the authors have considered the possibility of T. muris secretions/extracts inducing RELM-alpha on macrophages independently of Il-4R-alpha signalling.

Reviewer #3: 1. The authors main conclusion in this manuscript is that large intestinal macrophages can adopt an alternatively activated phenotype characterized by expression of CD206, RELMa and YM1 independently of IL4Ra signaling. However, their experimental approach leaves alternative interpretations on the table – the phenotype could be tissue specific or helminth specific. Can the authors assess whether Trichuris muris infection induces IL4Ra-dependent mac polarization in other tissues, like the peritoneal cavity (or others, as physiologically appropriate)? This experiment will test whether large intestinal cells are uniquely IL-4Ra-independent in this infection setting and would add important information to the field and provide data to support the comparisons to H polygyrus and Litomosoides infections in text associated with Figure 4. An additional comparison that could add value to the field would be evaluating the IL-4Ra-dependence of RELMa+ intestinal macrophages in response to a helminth that has previously been shown to require IL4Ra for alternative activation of macrophages in another tissue (if feasible).

2. Data presented in Supplemental Figure 2a demonstrates that IL4Ra expression is reduced on colonic macrophages isolated from T muris-infected Cre+ mice compared to Cre- controls, but doesn’t conclusively show that it is absent on the Cre+ cells. Authors have access to IL4Ra-/- mice (as evidenced by data in Figure 4). A comparison of the IL4Ra MFI on colonic macrophages between the Cre+ cells and cells isolated from IL4Ra-/- would be much more informative and would be necessary to support the claim of efficient and sustained deletion in the Cre+ animals.

**Part III – Minor Issues: Editorial and Data Presentation Modifications**

Reviewer #1: Minor comments:

- the use of the term ‘anti-inflammatory’ to characterize the macrophages present in the intestine during T. muris infection is probably not the best choice to describe the macrophage response in this context. There is an inflammation that develops against the presence of the worms in the large intestine, and although the inflammation is controlled and leads to repair mechanisms this is still ‘inflammation’. As the authors seem to consider important to use the correct terms with the description of M(IL-4) proposed in Murray et al. 2014; I would recommend to revisit their description of those ‘anti-inflammatory’ macrophages in the gut during worm infection.

- there is no page number nor line numbers, which makes the manuscript difficult to read and review.

- the term ‘large intestinal macrophages’ could be confusing. The adjective ‘large’ refers to the intestine, but could also refer to ‘intestinal macrophages’; as there are ‘large macrophages in the intestine’.

- overall, the choice of using the terms ‘Cre+’ vs ‘Cre-‘ in the results section and in the figures; while also using the full name IL4Rafl/fl.CX3CR1Cre is confusing. Cre+ is to me too general and renders difficult the reading and understanding of the results without referring to the results section.

- Results section 1 and FigS2: the gating strategies to analyze the selected macrophage population should be shown. Moreover, the results section states that CD206, RELMa and Ym1 are down in the CX3CR1-Cre+ mice, but FigS2c, d and e show results from CD206, PD-L2 and RELMa. Which markers have been tested?

- The results in FigS2a show MFI data for IL4Ra staining on gut macrophages at d35 post-infection, but panels b-e show bone-marrow macrophages. Even though there is a reduction in the MFI, this is not sufficient to assess the efficiency of IL4Ra impaired expression. Comparison with IL-4Ra-/- macrophages should be done, as well as a comparison between steady state and infected tissue (ideally including the main subpopulations of monocytes/macrophages – based on Ly6C and MHC-II expression (P1, P2, P3) and CD4/TIM-4)

- In Figs. 1, S2, 3d-f, 4g-h,j, 5f-h: The entire population of macrophages seems to shift with antibody staining. Thus, an additionnal parameter to analyze are the median fluorescence intensities which should be shown instead of % of positive cells (or both information). In Fig. S2, MFI was chosen for IL4Ra but % were chosen in figs S2 c-e.

- Fig. 3a: P1-P3 gates should be shown on the contour plots. Moreover, the TIM-4 and CD4 staining should be shown as FACS plots as well. In addition, the result section states that numbers of the macrophage subpopulations were affected upon infection, but the plotted data are percentages. Absolute cell numbers should be shown.

- results section: “contribution of both CD45.1 and CD45.2 cells to CD4+ and CD8+ T cell derived IFNγ and IL13 production” referring to figS3c-e is confusing to conclude that IL4Ra-/- cells produce amount of the cytokines similar to the WT CD45.1 compartment. The proportion of IFNG+ of IL-13+ CD4 or CD8 T cells in both CD45.1 and CD45.2 should be shown for clarity.

Reviewer #2: The work is of interested and includes relevant essays with suitable controls.

The authors should show the data supporting expressing of intracellular RELM-alpha in CD206 macrophages post Trichuris infection.

When analysing the gene expression of CD206 positive and negative macrophages (Supp Fig 1, a,b), the authors should stress that upregulation of Mrc1 in CD206 positive cells should be expected as this transcript encodes CD206. Indeed, this would be a nice validation of their sorting strategy.

The authors demonstrate that different macrophage populations (peritoneal, colon, liver) from IL4R-alpha fl/fl.CXC3Cr1Cre+ mice fail to upregulate RELM-alpha in response to IL-4 complexes injected in the peritoneal cavity. I might have missed it but could not find the source and amount of Il-4C injected. It is unclear what the authors refer to in legend of Figure 1 as (naïve; n=3-5, d18; n=6) as far as I can tell these animals were not infected. Also, I would like to enquire about their data presentation. All cells display some level of RELM-alpha expression based on the comparison of FMO control and specific labelling. Is this labelling real? If so it might be better to plot MFI rather than % of positive cells as at the moment it seems as most cells are positive and there is just upregulation of RELM-alpha expression.

Data from bone marrow derived macrophages (Supp Figure 2 c-e) complement findings with ex vivo macrophages. These validation results would be more complete if in addition to % of cells expressing markers authors show MFI. I was surprised by the relatively high expression of RELM alpha in unstimulated macrophages (15% of population already express it and only increases to 35% in response to IL-4) and I wonder if the authors would like to comment on that. In our experience CD206 is normally very well expressed by a high proportion of bone marrow derived. Based on the data less than 1-2% express CD206 and this % only increases to less than 10% in the presence of IL-4. Looking at their protocol, I did not notice any obvious reasons for these discrepancies. The authors included treatment with LPS and IFN-gamma, which as not been shown and collaction of sups that have not been characterised. Finally, I wonder if the Accutase treatment could account for the low levels of CD206 detected.

In figure 2 I would recommend including the label of the graphs in the actual figures as it is difficult to follow. Also, in my opinion it makes more sense to express Ab responses as titres rather than OD values. In addition, it seems that there is a consistent increase in IL-4 and IL-13 production in mLNs cultures from infected animals and I question whether the authors would like to comment on this.

When showing the different macrophage populations during infection in Figure 3 the authors might want to include the P1, P2 and P3 labels in the dot-blots to guide the reader. I would also recommend showing MFI in addition to % positive cells.

In Figure 4 I confess that I found it difficult to follow the discussion regarding the migration and maturation of IL-4R alpha positive and negative monocytes during infection. Also, the PDL-2 labelling seems to indicate the presence of a small population of CD45+ (WT) macrophages expressing PDL2 cells so it would be important to show the gating strategy to calculate % of PDL-2 positive cells.

Reviewer #3: 1. In the introduction, the authors state "This work is vital to inform strategies treating excessive, chronic intestinal inflammation as it highlights treatments which target the IL4R to promote anti-inflammatory macrophages are unlikely to be efficacious." This statement may not be entirely justified, as previous studies have demonstrated that IL-4 is sufficient to induce macrophage expression of RELMa, YM1, Arg1 and PDL2. Without a more broad characterization and accounting of functional differences between IL4-induced and Trichuris-induced but IL-4Ra-independent macrophages, authors are cautioned not to over-interpret the implications of their findings and to present them bearing in mind the difference between necessary and sufficient.

2. In the introduction, authors state “Many laboratories have demonstrated that these markers are also IL4Ra-dependent in vivo and these have become reliable indicators of M(IL4), especially in the context of type 2 inflammation and helminth infections.” Please provide some references for readers who are interested in comparing the current findings with existing literature. Additionally, since the helminth family is quite broad and infections have their own unique features, it could be helpful to specify which helminth infections have been shown to induce IL4Ra-dependent macrophage polarization.

3. Could not find Supplemental Figure Legends in the assembled PDF. Without those and other information, it’s not entirely clear what is being shown in Supp Fig 1. Can authors provide sort plots showing the sorted populations and/or include clarifying text: were macrophages and monocytes from P1, P2 and P3 included as a single population and sorted based solely on +/- expression of CD206? And was this only from T muris infected mice? What day post-infection? Finally, data shown in Supp Fig 1b (genes down-regulated in CD206+ vs CD206- mono/macs) is not mentioned in text.

4. Authors could consider annotating Fig 1 with source tissues (a, b: PEC; c, d: Colon; e, f: Liver). Similarly, in Fig 2b, annotating the days of infection would add clarity.

5. Text related to Supp Fig 2c-e indicates that CD206, RELMa and Ym1 were measured, but graphs shown quantify CD206, PDL2 and RELMa. Please revise for consistency. Additionally, can authors provide flow plots for transparency of what is being measured?

6. On Fig 3, can authors provide clarifying annotations eg P1, P2 and P3 on Fig 3a and a color-coded key for antigens stained for immunofluorescence (not IHC) images shown in Fig 3G.

7. Is there a disconnect between the justification of using CD206 for sorting mono/macs for RNA Seq (“Cells were sorted using FACS and isolated into two subsets based on surface expression of the marker CD206, chosen based on good co-expression with intracellular RELMα post Trichuris infection (data not shown)”) and the data that is shown in Fig 3? At day 18 post-Trichuris infection, there is no significant increase in CD206-expressing colonic macrophages (~40-50% at d0 and d18), but RELMa+ colonic macrophages increase from ~25% in naïve animals to ~50% at day 18 post-infection. Can the authors provide some clarification? Perhaps showing flow plots of CD206 x RELMa would help.

8. Can authors add % +/- SEM of the IL4Ra-/- and WT populations on the flow plots of blood and large intestinal monocytes in Fig 4c?

9. RELMa staining does not appear to differ between naïve and infected mice in Fig 4i, despite quantification in Fig 4G indicating a 10-15% increase. Overlays, annotation of RELMa MFI or added gates for spatial context may be helpful.

10. In the Methods section, reference to E/S products can be removed.

PLOS authors have the option to publish the peer review history of their article (what does this mean?). If published, this will include your full peer review and any attached files.

Reviewer #1: No

Reviewer #2: **Yes: **Luisa Martinez-Pomares

Reviewer #3: No
---

## [Decision Letter · Decision Letter 1]

29 Jun 2021

Dear Dr Forman,

We are pleased to inform you that your manuscript 'Trichuris muris infection drives cell-intrinsic IL4R alpha independent colonic RELMα+ macrophages.' has been provisionally accepted for publication in PLOS Pathogens.

Best regards,

Meera Goh Nair

Associate Editor

PLOS Pathogens

P'ng Loke

Section Editor

PLOS Pathogens

Kasturi Haldar

Editor-in-Chief

PLOS Pathogens

orcid.org/0000-0001-5065-158X

Michael Malim

Editor-in-Chief

PLOS Pathogens

orcid.org/0000-0002-7699-2064

In this revised submission, the authors have addressed the comments from all three reviewers.

Reviewer Comments (if any, and for reference):

Reviewer's Responses to Questions

**Part I - Summary**

Reviewer #1: The authors have addressed my comments thoroughly and I thank them for their response and revisions. The manuscript is overall well improved.

**Part II – Major Issues: Key Experiments Required for Acceptance**

Reviewer #1: no additional issue

**Part III – Minor Issues: Editorial and Data Presentation Modifications**

Reviewer #1: no additional issue

PLOS authors have the option to publish the peer review history of their article (what does this mean?). If published, this will include your full peer review and any attached files.

Reviewer #1: No

---

## [Editor Report · Acceptance letter]

22 Jul 2021

Dear Dr Forman,

We are delighted to inform you that your manuscript, "Trichuris muris infection drives cell-intrinsic IL4R alpha independent colonic RELMα+ macrophages.," has been formally accepted for publication in PLOS Pathogens.

Best regards,

Kasturi Haldar

Editor-in-Chief

PLOS Pathogens

orcid.org/0000-0001-5065-158X

Michael Malim

Editor-in-Chief

PLOS Pathogens

orcid.org/0000-0002-7699-2064